# Spatial distribution of ticks and tick-borne pathogens in central Hokkaido, Japan and associated ecological factors revealed by intensive short-term survey in 2024

**Mebuki Ito[1], Yuma Ohari[1,2], Mai Kishimoto[3,4], Keita Matsuno**[1,2,5,6]*

**1** Division of Risk Analysis and Management, International Institute for Zoonosis Control, Hokkaido University, Sapporo, Japan, **2** One Health Research Center, Hokkaido University, Sapporo, Japan, **3** Laboratory of Veterinary Microbiology, Graduate School of Veterinary Science, Osaka Metropolitan University, Izumisano, Japan, **4** Osaka International Research Center for Infectious Diseases, Osaka Metropolitan University, Osaka, Japan, **5** Institute for Vaccine Research and Development, HU-IVReD, Hokkaido University, Sapporo, Japan, **6** International Collaboration Unit, International Institute for Zoonosis Control, Hokkaido University, Sapporo 001, Japan

* matsuk@czc.hokudai.ac.jp

## Abstract

Tick-borne pathogens are transmitted by tick bites to cause infectious diseases in humans and domestic animals. To anticipate tick-borne disease occurrence, a high resolution understanding of infection risk distribution and its ecological drivers is needed. We aimed to map the spatial distribution of ticks and pathogen-infected ticks in central Hokkaido, Japan. Adult and nymphal ticks were collected with constant effort at 171 sites from 13 May to 26 June 2024, followed by screening tick-borne pathogens and ecological niche modeling. A total of 9,908 ticks were collected and the endemic tick-borne pathogens (i.e., tick-borne encephalitis virus, Yezo virus, Beiji nairovirus, Lyme disease group borreliae, and relapsing fever group borreliae) were primarily detected in *Ixodes* spp. ticks. Potential suitable habitats of the ticks and pathogens were predicted using the presence/absence data based on tick collection and pathogen detection. The models achieved acceptable predictive performance (median AUC = 0.83 and median TSS = 0.59 in leave-one-out cross-validation). *Ixodes persulcatus* and *Ixodes ovatus* were identified as the primary ticks for determining the distributions of all the pathogens. Besides, the predicted suitable habitats differed among pathogen and tick species. Among the environmental variables considered for modeling, snow depth appeared to significantly contribute to the distribution differences between ticks and pathogens. The findings of this study expand our understanding of the spatial risk distribution of tick-borne pathogen infections and its ecological context.

**Data availability statement:** The exact sampling locations cannot be publicly shared due to the ethical policy. The coordinate data will be made available on reasonable request to the corresponding author (KM, matsuk@czc.hokudai.ac.jp). Aggregate data are provided within the manuscript and supplemental information files (Supplementary Table S7). Sample R script is available on GitHub at: https://github.com/DRAM-IIZC/MebukiITO_Riskmap.

**Funding:** This work was supported by the Japan Agency for Medical Research and Development (AMED; https://www.amed.go.jp/en/), Japan, under grant numbers JP23jf0126002 and JP25fk0108717 (K.M.); by the Ministry of Education, Culture, Sports, Science and Technology (MEXT; https://www.mext.go.jp/en/)/Japan Society for the Promotion of Science (JSPS; https://www.jsps.go.jp/english/) KAKENHI, Japan, under the grant numbers JP23K27066, JP23K20041, JP26H02351 (K.M.), JP23K14090, JP26K09121 (Y.O.), and JP25KJ0451 (M.I.); by AMED SCARDA World-leading institutes for vaccine research and development Hokkaido Synergy Campus (https://www.amed.go.jp/en/program/list/21/02/002.html) under the grant number JP223fa627005 (K.M.); by Support for Pioneering Research Initiated by the Next Generation of Japan Science and Technology Agency (JST SPRING; https://sites.google.com/eis.hokudai.ac.jp/exexphd-fellow/home) under grant number JPMJSP2119 (M.I.); and by The Chemo-Sero-Therapeutic Research Institute (KAKETSUKEN). The funders had no role in study design, data collection and analysis, decision to publish, or preparation of the manuscript.

**Competing interests:** The authors declare no competing interests.

## Introduction

Tick-borne pathogens associated with ixodid ticks (Acari, Ixodidae) are generally maintained in the field by ticks and wild animals and infect humans and domestic animals via tick bites to cause diseases [1–3]. Ixodid ticks are reservoirs of tick-borne pathogens, which are responsible for harboring them for long periods and transmitting to animals [4,5]; the pathogens can be retained over multiple developmental stages of ticks (i.e., trans-stadial transmission) and some of them can be transmitted through eggs (i.e., transovarial transmission). Naïve ticks are infected with tick-borne pathogens through blood by feeding on infected animals [4,5] or by directly acquiring pathogens from infected ticks simultaneously feeding on the same host (i.e., co-feeding transmission) [6]. In the natural lifecycle of tick-borne pathogens, wild animals also play pivotal roles not only by amplifying pathogens for transmission to ticks as reservoir hosts but also serving as hosts of ticks themselves [7]. Thus, understanding of the associations among pathogens, ticks, and wild animals are essential to clarify circulation of tick-borne pathogens in nature and to anticipate the occurrence of tick-borne diseases.

Each tick-borne pathogen has its own vector tick species and preferential animal host species, which are capable to maintain, amplify, and transmit the pathogen. For example, tick-borne encephalitis virus (TBEV) is mostly transmitted by ticks of *Ixodes ricinus* species complex, and small rodents are essential animal hosts for the circulation of TBEV [1]. While the specific vector ticks and animal host species are directly associated with the lifecycle of tick-borne pathogens, the association between certain tick and non-reservoir hosts indirectly affecting the pathogen lifecycle is not negligible [7]. Tick-host association is particularly influential in the spread of tick-borne pathogens; wild animals capable of long-distance dispersal, such as birds, can carry infected ticks to the outside of their endemic areas [8,9]. Furthermore, since lives of ticks and wild animals rely on the environmental resources, the circulation of tick-borne pathogens is indirectly affected by the environmental factors, such as landscape, climate, and topography [10,11]. These direct and indirect interactions among pathogens, ticks, wild animals, and the environment influence the occurrence of tick-borne diseases.

Hokkaido is the northernmost island prefecture in Japan with a distinct climatic and biological characteristics from the main island of Japan (Honshu) and shares ticks and tick-borne pathogens with the northeastern Eurasian continent. In this region, five tick-borne pathogens associated with the *Ixodes* ticks have caused or potentially cause human diseases: TBEV, Yezo virus (YEZV), Beiji nairovirus (BJNV), Lyme disease group borreliae (LDB), and relapsing fever group borreliae (RFB) (i.e., *Borrelia miyamotoi*) [1,12–19]. Tick-borne encephalitis, Lyme disease, and relapsing fever are notifiable diseases in Japan, with 2, 18, and 11 cases reported in Hokkaido in 2024, respectively [20]. Emerging tick-borne virus infections caused by YEZV have been found in 22 cases in Hokkaido since the first report in 2021 [13,15,16,21,22]. No human cases of BJNV infection in Japan have been reported, while BJNV is widely distributed in *Ixodes* ticks in Hokkaido [23]. While adult *Ixodes persulcatus* and *Ixodes ovatus* cause almost all tick bite cases in Hokkaido [24,25], *Haemaphysalis*

ticks are also distributed [26], e.g., *Haemaphysalis longicornis*, which is a vector tick species to transmit severe fever with thrombocytopenia syndrome [27]. Although previous studies provided valuable insights into these pathogens and *Ixodes* spp. ticks, their geographical coverage has been patchy [12,23,28–30]. Consequently, a high-resolution understanding of their respective spatial distributions is still lacking. This knowledge gap hinders an understanding of each tick/pathogen's ecology and effective risk assessment of tick-borne diseases in this region. To address this gap, spatially comprehensive and comparable analysis across tick/pathogen species is required.

Here, we aim to examine the environmental determinants of distributions of ticks and tick-borne pathogens in central Hokkaido, one of the most populated regions in Japan. Based on the comparable data between ticks and pathogens detected in the ticks obtained by constant effort sampling at 171 sites in 1.5 months and screening of five pathogens, we performed the ecological niche modeling (ENM) [31] to predict the potential suitable habitats of ticks and tick-borne pathogens by using landscape, climate, topography, and wild mammal distributions as explanatory variables. Further, we examined the distribution differences among tick/pathogen species by quantifying the spatial clustering of each species distribution and the equivalency between the spatial niches of each tick and each pathogen. Our findings contribute to the identification of ecological factors associated with spatial distribution of ticks and tick-borne pathogens as well as consideration of preventive measures against tick-borne diseases in the study area.

## Methods

### Tick sampling

The study area (44°00.0′N – 42°63.3′N, 142°02.5′E – 140°91.25′E) is at central Hokkaido, Japan (Fig 1) with an east-west length of 124 km and a north-south length of 209 km, approximately. The study area is centered on Sapporo city, which has a population of approximately two million (https://www.city.sapporo.jp/toukei/jinko/jinko.html, accessed February 5, 2025). The plain region encompassing Sapporo city is surrounded by three mountainous regions, which are referred to in this study as northern, southern, and eastern mountains (Fig 1).

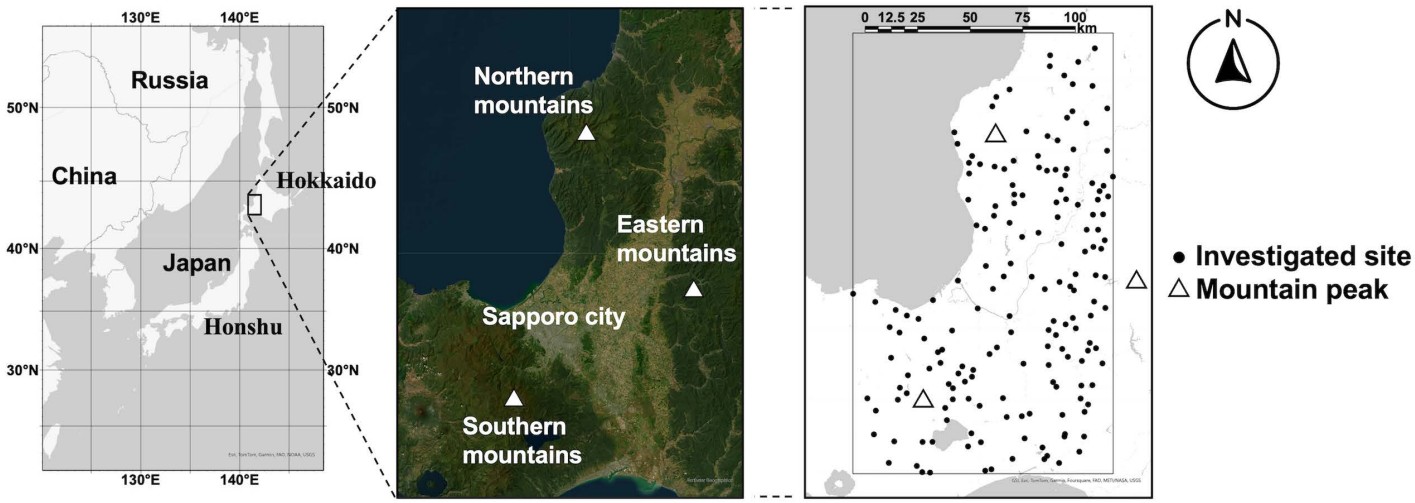

**Fig 1. Study area.** Hokkaido is the northernmost island prefecture of Japan, distinct from the main island of Japan (Honshu). At the center of Hokkaido, our study area (indicated by the black square in the right panel) is located. Here, mountainous regions surround Sapporo city, one of the largest cities in Japan. The representative peak of each mountainous region (referred as northern, eastern, or southern mountains) is marked as a triangle. Each tick collection site is represented by a black dot in the right panel. The satellite image in the middle is also shown in Fig 2 and 3. Basemap images were provided as Esri's ArcGIS basemaps. Content is the intellectual property of Esri and is used herein with permission. Copy right © 2025 Esri and its licensors. All rights reserved.

By flagging method using a 0.70 m × 1.00 m flannel fabric, adult and nymphal ticks were collected at randomly selected 171 sites (black dots in the right panel of Fig 1) in 2024. To confirm that the investigated sites were randomly located across our study area, an average nearest neighbor analysis was performed in Arc GIS pro ver. 3.3.2 [32]. The study period was set from 13 May to 26 June, aligning with the phenology of ticks in our study area [24,28]. All collections were conducted during daylight hours on days with no precipitation. Tick collection in the present study was performed only by the first author (M.I.) for 30 minute at each site to maintain constant sampling effort. We selected the time-based sampling effort since it was effective to standardize the sampling effort across diverse habitat types characterized by various vegetation heights and densities in our study area. All the investigated sites were publicly accessible places where no specific permissions are required for collecting arthropods, and this study did not involve endangered or protected species. The species and sex of adult ticks and genus of nymphal ticks were identified under a stereomicroscope based on morphological features [26,33]. Adult ticks ($\leq$ 10 individuals) were pooled per species, sex, and site, and nymphal ticks ($\leq$ 100 individuals) were pooled per genus and site. Ticks which died during transportation to the laboratory were not used. Each pool was transferred into a taco™Prep Pre-loaded Steel Bead Tube (GeneReach Biotechnology Corp., Taichung City, Taiwan), followed by washing once with distilled water. The pooled ticks were stored at –80°C until nucleic acid extraction.

## Nucleic acid extraction and pathogen detection

From the pooled ticks, DNA and RNA were extracted using the blackPREP tick DNA/RNA Kit (Analytikjena, Jena, Germany) following the manufacturer's instructions except that ticks were homogenized with 100 μL of Solution RL using Micro Smash MS-100R (TOMY, Tokyo, Japan) three times at 3,000 rpm for 30 s at 4°C. Extracted DNA and RNA were stored at –80°C until further experiments. Genome detection of five endemic tick-borne pathogens in the study area was performed using reverse-transcription quantitative real-time polymerase chain reaction (RT-qPCR) or qPCR on a qTOEWR³ G Real-Time PCR Thermal Cycler (Analytikjena, Jena, Germany) as described below.

TBEV RNA was detected according to Achazi et al. (2011) [34] using One Step PrimeScript III RT-qPCR mix (Takara Bio Inc., Shiga, Japan) with primer 1: 5′–TGGAYTTYAGACAGGAAYCAACACA–3′, primer 2: 5′– TCCAGAGACTYTGRT-CDGTGTGGA–3′, and probe: 5′–FAM–CCCATCACTCCWGTGTCAC–MGB–NFQ–3′ targeting the NS1 gene, under the following thermal conditions: 50°C for 15 minute and 95°C for 2 minute followed by 45 cycles of 95°C for 15 s and 60°C for 34 s.

YEZV RNA was detected using One Step PrimeScript III RT-qPCR mix (Takara Bio Inc., Shiga, Japan) with primer 1: 5′–CATACAGGAAGGCCATCTCATT-3′, primer 2: 5′–AGCCCTTGACACTGCATTT–3′, and probe: 5′–/56–FAM/ACCTAC-TAC/ZEN/TGGATGTGGAAGGCAGA/3IABkFQ/–3′ targeting the N gene, under the following thermal conditions: 52°C for 5 minute and 95°C for 10 s followed by 40 cycles of 95°C for 5 s and 60°C for 30 s.

BJNV RNA was detected targeting the N gene with One Step PrimeScript III RT-qPCR mix (Takara Bio Inc., Shiga, Japan), primer 1: 5′– ATGGCTCATGAACGCAAC-3′, primer 2: 5′– GGTYTTGATGACYTCTGGGTC–3′, and probe: 5′–/56-FAM/AAAGCTCTGACCGTTTACCCAGCC/3IABkFQ/–3′, where locked nucleic acid is underlined. The locked nucleic acids were introduced for distinguishing BJNV from the similar nairoviruses, such as Yichun nairovirus, that have not been reported as human pathogens. The thermal conditions were as follows: 52°C for 5 minute and 95°C for 10 s followed by 45 cycles of 95°C for 5 s and 60°C for 30 s.

LDB and RFB genome DNAs were detected by multiplex qPCR according to Barbour et al. (2009) [35] and Takano et al. (2014) [12] using Premix Ex Taq (Probe qPCR) (Takara) with primer 1: 5′–GCTGTAAACGATGCACACTTGGT–3′, primer 2: 5′–GGCGGCACACTTAACACGTTAG–3′, probe for LDB: 5′–6FAM–TTCGGTACTAACTTTTAGTTAA–3′, and probe for RFB: 5′–VIC–CGGTACTAACCTTTCGATTA–3′ targeting the 16 S rRNA gene, under the following thermal conditions: 95°C for 20 s followed by 45 cycles of 95°C for 5 s and 60°C for 30 s.

All assays were performed in technical duplicate with positive and no-template negative controls. Samples with Ct values < 45 showing exponential amplification curves in both of duplicated assays without amplifications in the corresponding

negative control were considered positive in each RT-qPCR and qPCR. For samples exhibiting exponential amplification in only one replicate, the assay was repeated in technical duplicate; only those showing reproducible amplification in both of the runs were strictly classified as positive. The positive rate (%) of each pathogen per tick and its 95% confidence interval (95% CI) were estimated by maximum likelihood using the PooledInfRate package in R [36] based on the results of genome detection from pooled ticks.

LDB is also known as *Borrelia burgdorferi* sensu lato (s.l.), which is divided into several genospecies carried by specific tick species [2,3]. In our study area, the associations between the genotypes of LDB and tick species have been well confirmed by multiple studies [2,5,18,37]. Furthermore, in Japan, most Lyme disease cases in humans are caused by the genospecies of LDB specifically carried by *I. persulcatus* (i.e., *Borrelia garinii* and *Borrelia afzelii*) [5,18,37]. Thus, in subsequent steps, LDB detected from *I. persulcatus* was referred to as pLDB, which was analyzed separately in addition to the total LDB.

### Definition of presence/absence of each tick/pathogen at a site

To use tick collection data and pathogen detection data for subsequent spatial analysis and modeling, those data at each investigated site were converted to presence or absence using the definitions: tick present if ≥1 adult collected; pathogen present if ≥1 positive pool; all other sites were absent sites. Adults were used exclusively for this definition for ticks to ensure the highest accuracy of morphological identification.

### Spatial clustering of each tick and pathogen species

For investigating spatial clustering of each tick and pathogen species, Moran's *I* test was performed under the alternative hypothesis of positive spatial autocorrelation using the spdep package in R [38]. Moran's *I* was calculated as follows [39]:

$$I = \frac{N}{\sum_i \sum_j w_{ij}} \frac{\sum_i \sum_j w_{ij} (y_j - \bar{y})(y_i - \bar{y})}{\sum_i (y_i - \bar{y})^2}.$$

Here, $y_i$ represents the number of collected adults of each tick species or presence (1)/ absence (0) of each pathogen species at site $i$ and $\bar{y}$ is the mean of the variable. $N$ is the number of investigated sites, which is 171 in the present study. $w_{ij}$ is an element of a row-standardized spatial weights matrix, defined as:

$$w_{ij} = \exp(-d_{ij}/r)$$

where $d_{ij}$ represents the distance (km) between site $i$ and $j$. The parameter controlling the distance decay ($r$) was varied from 0.1 to 20 to examine the spatial autocorrelation at different spatial scales (Supplementary S1 Fig). Moran's *I* close to −1 means that neighboring sites are different from each other, while Moran's *I* close to 1 means that the species distribute in cluster. Moran's *I* close to 0 represents a random pattern. A p-value < 0.05 was considered statistically significant.

Following the global spatial analysis, the local spatial statistic G-star was calculated to analyze local clustering of each tick and pathogen using the spdep package [38]. We used the number of collected adults of each tick species or presence (1)/ absence (0) of each pathogen species at each site and the same spatial weights matrix as that used for Moran's *I*. Z-values were mapped using the ggplot2 package [40] to visualize hotspots of each species.

### Ecological niche modeling

For explanatory variables in ENM, 20 variables of landscape, climate, topography, and wild mammal distributions were included (Table 1 and Supplementary S2 Fig). Six landscape variables (i.e., the size of forest area, *forest*; grassland area, *grass*; wetland area, *wet*; cropland area, *crop*; urban area, *build*; and water area, *water*) were extracted from the

**Table 1. Potential explanatory variables considered to be included in ENM.**

| Variable | Explanation |
|----------|-------------|
| forest | Size of forest area (km²) |
| grass | Size of grassland area (km²) |
| wet | Size of wetland area (km²) |
| crop | Size of cropland area (km²) |
| build | Size of urban area (km²) |
| water | Size of water area (km²) |
| snow | Yearly maximum snow depth (m) |
| prec6* | Precipitation in June (mm) |
| temp6* | Average temperature in June (°C) |
| suns | Daylight hours in June (h) |
| cold | Lowest temperature in January (°C) |
| elev* | Average elevation (m) |
| angl* | Maximum slope angle |
| bear | Distribution of brown bear *U. arctos yesoensis* (presence or absence) |
| tanuki | Distribution of tanuki *N. procyonoides albus* (presence or absence) |
| fox | Distribution of fox *V. vulpes schrencki* (presence or absence) |
| raccoon | Distribution of raccoon *P. lotor* (presence or absence) |
| deer | Distribution of sika deer *C. nippon yesoensis* (presence or absence) |
| lon | Longitude |
| lat | Latitude |

* The squared terms of *prec6*, *temp6*, *elev*, and *angl* were also considered as explanatory variables.

High–Resolution Land Use Land Cover Map ver. 23.12 with an original resolution of 10 × 10 m provided by JAXA (https://www.eorc.jaxa.jp/ALOS/lulc/data/index_j.htm#japan_v23.12, accessed July 8, 2024). These landscape categories were classified based on observations from artificial satellites by JAXA. Five climatic variables (i.e., the yearly maximum snow depth, *snow*; the precipitation in June, *prec6*; the average temperature in June, *temp6*; the daylight hours in June, *suns*; and the lowest temperature in January, *cold*) and two topographic variables (i.e., the average elevation, *elev* and the maximum slope angle, *angl*) were sourced from the National Land Numerical Data with an original resolution of 1 × 1 km provided by Ministry of Land, Infrastructure, Transport, and Tourism, Japan (https://nlftp.mlit.go.jp/ksj/index.html, accessed July 8, 2024). These climatic values were the mean based on 30 years of meteorological observations by Japan Meteorological Agency and the topographic data was based on digital elevation model (DEM) created by the Geospatial Information Authority of Japan. The distribution information of mammals were obtained from the Distribution survey of birds and mammals with special attention and the Medium- and large-sized mammal distribution survey provided by Ministry of the Environment, Japanhttps://www.biodic.go.jp/youchui/youchui_top.html#mainText and https://www.biodic.go.jp/kiso/do_kiso4_mam_f.html#mainText, respectively, both accessed July 26, 2024), the Raccoon capture information 2020(R2) provided by Hokkaido government (https://www.harp.lg.jp/opendata/dataset/1742.html, accessed July 26, 2024), and the Information map on hunting sika deer provided by Hokkaido Research Organization (https://www.hro.or.jp/industrial/research/eeg/development/datamap/deermap.html, accessed July 26, 2024). These mammal maps were provided in the same 5 × 5 km grid. For each mammal species, we defined a present grid as a grid which one or more data sources treated as a present grid, while the other grids were defined as absent grids (the distribution of brown bear *Ursus arctos yesoensis*, *bear*; the distribution of tanuki *Nyctereutes procyonoides albus*, *tanuki*; the distribution of fox *Vulpes vulpes schrencki*, *fox*; the distribution of raccoon *Procyon lotor*, *raccoon*, and the distribution of sika deer *Cervus nippon*

*yesoensis*, deer). All the climatic and topographic variables were provided in the same 1 × 1 km grid. The landscape variables, which were provided in 10 × 10 m grids, were aggregated to the 1 × 1 km grid. For the mammal variables, the data from each 5 × 5 km grid was assigned uniformly to all 25 of the 1 × 1 km grids nested within it. Although the 5-km grid is the highest resolution available for mammal distribution data, uniform downscaling to 1 km might overlook fine-scale variations. ArcGIS Pro ver. 3.3.2 [31] was used to extract the explanatory variables.

Ecological niche modeling was performed using generalized linear model (GLM) with a resolution of 1 × 1 km. We used the stats package in R [41] for fitting GLM. As a response variable, presence/absence of each tick/pathogen species was used, assuming that it followed a binomial distribution (using a logit link function). Each continuous explanatory variable was standardized by subtracting its mean and dividing by its standard deviation. To avoid model complexity and preserve statistical power, no interaction terms between environmental variables were considered. For variable selection, we developed the best prediction model through a two-step process: an automated selection with Akaike Information Criteria (AIC) followed by a manual selection. As an initial step to identify a candidate set of variables, we selected the AIC-best model from all combinations of explanatory variables using the MuMIn package in R [42]. Starting with the variables from this AIC-best model, a manual variable selection was performed as described below, considering both prediction accuracy and model complexity. Significant explanatory variables (p-value < 0.05) in a single regression analysis were prioritized to be included in the model. The explanatory variables to be included in the model were selected using manual forward and backward stepwise selection based on the area under curve of the receiver operating characteristic (AUC) [43] and the true skill statistic (TSS) [44] calculated using leave-one-out cross-validation (LOOCV) [31]. AUC ranges from 0 to 1, with > 0.7 being an acceptable threshold for predictions, and TSS ranges from –1–1, with > 0.5 being an acceptable threshold for predictions. To avoid multicollinearity, variance inflation factors (VIFs) were calculated using the performance package [45]. When the VIFs exceeded 5, correlated explanatory variables were excluded one by one and the variable combination with the best prediction accuracy (i.e., more AUC and TSS) was selected. We also explored spatial autocorrelation in the residuals of each model using Moran's *I* test as mentioned above. When spatial autocorrelation was detected in the residuals of GLM, we fitted a generalized additive model to apply spatial smoothing of the grid coordinate (longitude, *lon* and latitude, *lat*), using the mgcv package in R [46]. We considered a candidate set of smooth terms based on thin plate regression splines: one-dimensional smooths of longitude (s(*lon*)) and latitude (s(*lat*)) for large-scale trends, and a tensor product interaction (ti(*lon*, *lat*)) for more localized patterns. Starting with the non-spatial model, we incrementally added these terms and selected the simplest model for each species until the residuals showed no remaining significant spatial autocorrelation. After the best model was selected, we performed a stratified 5-fold cross-validation to evaluate the predictive performance of the best model. This method was chosen to maintain the balanced ratio of presence/absence in each fold. As performance metrics, we calculated AUC/TSS and reported the mean and standard deviation (SD) of these metrics across the five folds for each tick/pathogen. Finally, the best model for each tick/pathogen species was used to predict the presence probability of each grid, which was then mapped across the study area. The predicted probability was converted to binary format using the threshold that maximized TSS to map the potential distribution in presence/absence format, as well. For the visualization, the ggplot2 package in R [40] was used.

## Spatial overlap between pathogen and tick

To quantify spatial overlap between each pathogen species and each tick species, Schoener's *D* and Warren's *I* were calculated. Both of the metrics measure how equally two species use space relative to its availability [47–49].

Schoener's *D* is defined as

$$D(p_X, \ p_Y) = 1 - \frac{1}{2} \sum_i \left| p_{X,i} - p_{Y,i} \right|,$$

while Warren's *I* is defined as

$$I\left(p_X,\ p_Y\right) = 1 - \frac{1}{2}\sum_i \left(\sqrt{p_{X,i}} - \sqrt{p_{Y,i}}\right)^2.$$

In the above equations, $p_{X,i}$ denotes the presence probability of grid *i* predicted by the ENM for species X. Both Schoener's *D* and Warren's *I* range from 0 (no overlap between two species) to 1 (complete overlap). To calculate 95% confidence intervals, we employed a bootstrap approach with 1,000 replicates.

## Results

### Prevalence of ticks and tick-borne pathogens in central Hokkaido, Japan

An average nearest neighbor analysis suggested that the 171 investigated sites were located randomly (z-score = −0.502 and p-value = 0.616).

One or more ticks were collected at 159 of the 171 investigated sites, comprising a total of 4,608 adult ticks and 5,300 nymphal ticks (Table 2, Fig 2, and Supplementary S1 Table). Adult ticks were identified as nine species: *I. ovatus, I. persulcatus, Ixodes pavlovskyi, Ixodes tanuki, Haemaphysalis megaspinosa, H. longicornis, Haemaphysalis flava, Haemaphysalis japonica,* and *Haemaphysalis concinna. Ixodes ovatus* and *I. persulcatus* were the most common species, accounting for 82% and 11% of collected adult ticks, respectively. Only one adult of each *I. tanuki* and *H. concinna* was collected; consequently, these tick species were excluded from the spatial analyses. A total of 9,745 live ticks were pooled into 1,026 samples and used for RT-qPCR and qPCR (Table 2). TBEV, YEZV, BJNV, LDB, and RFB were detected in seven (mean Ct values: 23.4–38.3), eight (mean Ct values: 24.2–36.2), 11 (mean Ct values: 15.6–26.9), 476 (mean Ct values: 30.6–42.0), and 16 (mean Ct values: 26.8–38.4) samples, respectively (Supplementary S2 Table). The detailed justification of our screening methods and results of the additional molecular confirmation, including sequencing, of screening-positive samples are provided in Supplementary S1 Text1. All the five pathogens were essentially detected in

**Table 2. Collected ticks and pathogens detected in the ticks.**

| Stage | Species | Collected ticks | Pooled ticks[1] | Median of ticks per pool | Number of pools | Number of positive pools | | | | | Prevalence in collected ticks (%)[2] | | | | |
|---|---|---|---|---|---|---|---|---|---|---|---|---|---|---|---|
| | | | | | | TBEV | YEZV | BJNV | LDB | RFB | TBEV | YEZV | BJNV | LDB | RFB |
| Adult | *I. ovatus* | 3780 | 3731 | 8 | 523 | 5 | 6 | 1 | 397 | 7 | 0.1 | 0.2 | 0 | 20.5 | 0.2 |
| | *I. persulcatus* | 494 | 489 | 2 | 152 | 1 | 1 | 9 | 65 | 6 | 0.2 | 0.2 | 1.9 | 19.6 | 1.3 |
| | *I. pavlovskyi* | 15 | 15 | 1 | 11 | 0 | 0 | 0 | 3 | 1 | 0 | 0 | 0 | 23.1 | 6.5 |
| | *H. megaspinosa* | 138 | 136 | 1 | 65 | 0 | 0 | 0 | 1 | 0 | 0 | 0 | 0 | 0.7 | 0 |
| | *H. longicornis* | 94 | 94 | 2 | 26 | 0 | 0 | 0 | 0 | 0 | 0 | 0 | 0 | 1.6 | 0 |
| | *H. flava* | 61 | 61 | 1 | 50 | 0 | 0 | 0 | 1 | 0 | 0 | 0 | 0 | 0 | 0 |
| | *H. japonica* | 24 | 24 | 2 | 13 | 0 | 0 | 0 | 0 | 0 | 0 | 0 | 0 | 0 | 0 |
| | Others[3] | 2 | 2 | 1 | 2 | 0 | 0 | 0 | 0 | 0 | 0 | 0 | 0 | 0 | 0 |
| Nymph | *Ixodes* spp. | 84 | 79 | 2 | 35 | 0 | 1 | 1 | 6 | 2 | 0 | 1.2 | 1.3 | 8.5 | 2.6 |
| | *Haemaphysalis* spp. | 5216 | 5114 | 19 | 149 | 1 | 0 | 0 | 3 | 0 | 0 | 0 | 0 | 0.1 | 0 |
| Total | | | 9908 | 9745 | 7 | 1026 | 7 | 8 | 11 | 476 | 16 | 0.1 | 0.1 | 0.1 | 5.9 | 0.2 |

[1]Ticks died during transportation to the laboratory were removed.

[2]Prevalence was estimated by maximum likelihood using the PooledInfRate package in R [36] based on the results of genome detection from pooled ticks.

[3]One *I. tanuki* and one *H. concinna*.

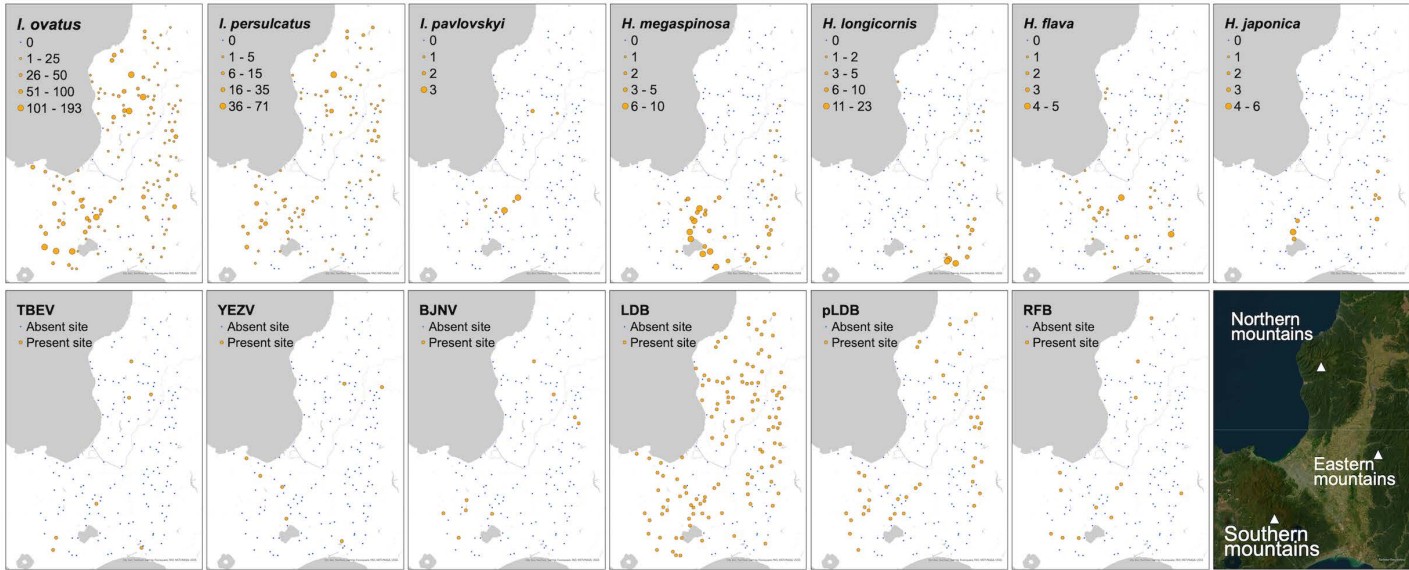

**Fig 2. Tick abundance and pathogen presence.** As for seven tick species used for spatial analyses, the number of collected adult ticks at each investigated site is represented with the size of circle. Present and absent sites of each pathogen species are represented with orange and blue dots, respectively. The satellite image on the bottom right is also shown in Fig 1 and 3. Basemap images were provided as Esri's ArcGIS basemaps. Content is the intellectual property of Esri and is used herein with permission. Copy right © 2025 Esri and its licensors. All rights reserved.

*Ixodes* spp. ticks. The prevalences of TBEV and YEZV were similar for *I. ovatus* and *I. persulcatus.* BJNV was detected almost exclusively in *I. persulcatus*, with its significantly higher percentage (1.92, 95% CI: 0.95–3.50) compared to that of *I. ovatus* (0.03, 95% CI: 0.00–0.13). The prevalence of LDB was comparable between *I. ovatus*, *I. persulcatus*, and *I. pavlovskyi*. Since most LD human cases in Hokkaido are caused by the genospecies of LDB specifically carried by *I. persulcatus* [2,18,36], LDB detected in *I. persulcatus* was separately analyzed as pLDB in the subsequent steps in addition to the total LDB. As for RFB, estimated positive rate in *I. persulcatus* (1.27, 95% CI: 0.52–2.64) and *I. pavlovskyi* (6.53, 95% CI: 0.39–27.28) were significantly higher than that in *I. ovatus* (0.19, 95% CI: 0.08–0.37).

The results of tick collection and pathogen detection were transformed to presence/absence data (Supplementary S3 Table). Notably, although the total number of collected *H. flava* was lower than that of *H. longicornis* (Table 2), the number of present sites for *H. flava* was higher than that for *H. longicornis*.

To examine the spatial clustering of the distributions of ticks and pathogens, Moran's *I* test was performed using the number of collected adults of each tick species and presence/absence data of each pathogen species (Table 3 and Supplementary Table S4). Independent of the distance decay parameter ($r$), Moran's *I* of all tick species, except for *H. flava*, were significantly positive. Moran's *I* for *H. flava* was not statistically significant with stronger distance decay. Moran's *I* for all pathogen species, except for YEZV, were significantly positive with at least either strong or weak parameter settings. Spatial clustering of YEZV was not detected using any setting of the parameter (Supplementary Table S4).

Moran's *I* and p-value were calculated using the spdep package in R [38] with the parameter controlling the distance decay ($r = 1$ or $r = 10$ as representatives of strong or weak distance decay, respectively). Full results ($r = 0.1$ to $r = 20$) are shown in Supplementary S4 Table.

To reveal hotspots of each tick and pathogen, the local spatial statistic G-star was calculated and visualized in Supplementary S3 Fig. The analysis identified hotspots that largely aligned with the raw tick collection and pathogen detection data mapped in Fig 2. *Ixodes* spp. ticks and associated pathogens like LDB exhibited prominent clusters in northern and

**Table 3. Results of Moran's *I* test to investigate spatial clustering of each tick/pathogen species.**

|  | Species | *r* = 1 |  | *r* = 10 |  |
|---|---|---|---|---|---|
|  |  | Moran's *I* | p-value | Moran's *I* | p-value |
| Tick | *I. ovatus* | 0.28 | <0.001 | 0.14 | <0.001 |
|  | *I. persulcatus* | 0.17 | 0.004 | 0.09 | <0.001 |
|  | *I. pavlovskyi* | 0.21 | 0.001 | 0.07 | <0.001 |
|  | *H. megaspinosa* | 0.51 | <0.001 | 0.26 | <0.001 |
|  | *H. longicornis* | 0.94 | <0.001 | 0.27 | <0.001 |
|  | *H. flava* | 0.10 | 0.083 | 0.09 | <0.001 |
|  | *H. japonica* | 0.38 | <0.001 | 0.06 | <0.001 |
| Pathogen | TBEV | 0.16 | 0.016 | 0.01 | 0.205 |
|  | YEZV | −0.05 | 0.736 | 0.00 | 0.427 |
|  | BJNV | 0.04 | 0.279 | 0.04 | 0.006 |
|  | LDB | 0.35 | <0.001 | 0.21 | <0.001 |
|  | pLDB | 0.40 | <0.001 | 0.12 | <0.001 |
|  | RFB | 0.20 | 0.003 | 0.03 | 0.041 |

southern mountains. We found that *H. flava* and YEZV, both of which were considered to have relatively dispersed distribution compared to other species, exhibit local clusters.

### Model performance and explanatory variables in ENM

LOOCV was performed to validate model performance. The best model for most tick/pathogen species achieved acceptable thresholds for predictions (AUC > 0.7 and TSS > 0.5). The model for *H. flava* exhibited poor predictive performance (AUC = 0.71 and TSS = 0.41) and did not meet the TSS threshold (Supplementary S5 Table). Overall, the median of AUC was 0.83 (Interquartile range, 0.76–0.86), and the median of TSS was 0.59 (Interquartile range, 0.53–0.64). In addition, a stratified 5-fold cross-validation was performed to evaluate the stability of the best model's predictive performance. As for all species, except for *H. flava*, the means of AUC and TSS were above acceptable thresholds for predictions (AUC > 0.7 and TSS > 0.5) and their standard deviations were small (Supplementary S5 Table). AUC and TSS for *H. flava* were below acceptable thresholds for predictions. Explanatory variables included in the best models are shown in Fig 3 and Supplementary S6 Table. The yearly maximum snow depth (*snow*) was included in the best models of all tick species, except for *I. pavlovskyi*, with a positive coefficient for *Ixodes* ticks and a negative coefficient for *Haemaphysalis* ticks. The elevation (*elev*) tended to be included in the best models of both tick and pathogen with a positive coefficient. As for landscape variables, the size of forest area (*forest*) with a positive coefficient was included in the best models of *I. ovatus*, *I. persulcatus*, *H. flava*, YEZV, and LDB, while the size of grassland area (*grass*) with a negative coefficient was included in the best models of *I. pavlovskyi*, *H. megaspinosa*, and pLDB. The distributions of mammals tended not to be included in the best models compared to landscape, climatic, and topographic variables. Spatial smoothing was applied for the model of *I. pavlovskyi*, *H. megaspinosa*, *H. longicornis*, *H. flava*, *H. japonica*, pLDB, and RFB (Supplementary S6 Table). No significant spatial autocorrelation (Moran's *I* test, p > 0.05) was detected in the residuals of the final models (Supplementary S5 Table).

### Predicted suitable habitats and spatial overlap

The presence probability of each tick/pathogen species at each grid predicted by the best model is mapped in Fig. 4. The potential distributions in binary (presence/absence) format are mapped in Supplementary S3 Fig. The predicted suitable habitats for *Ixodes* ticks were at the whole mountainous regions except for the southern part of the study area. The

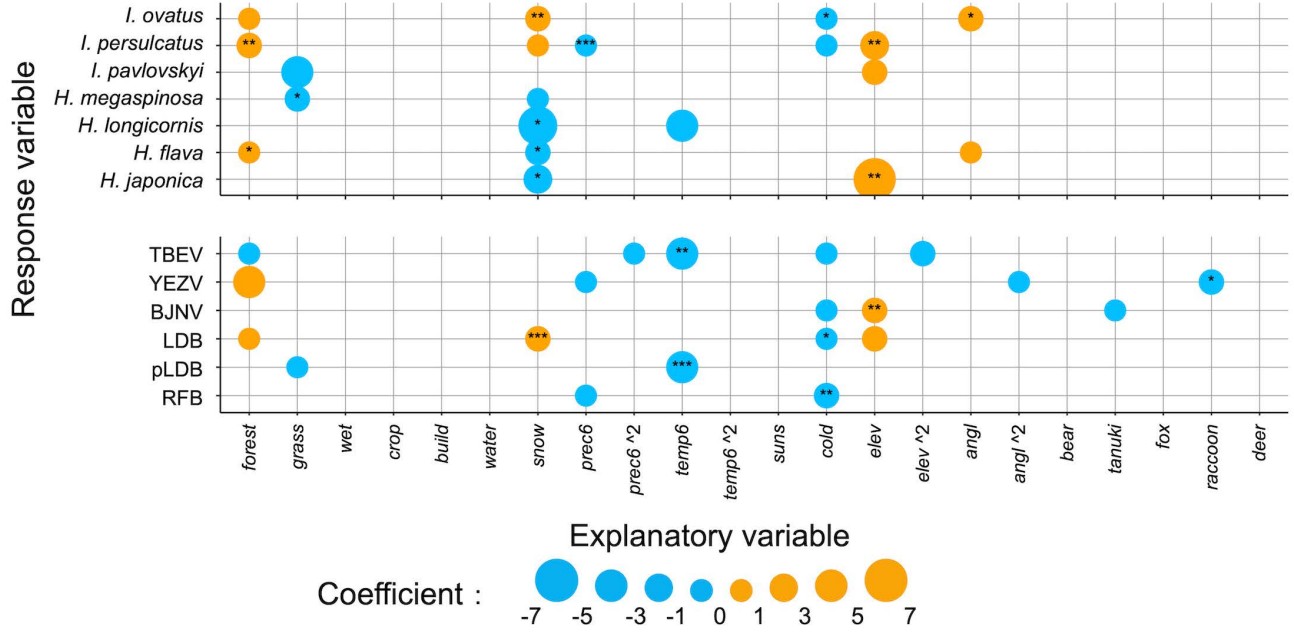

**Fig 3. Explanatory variables included in the best models.** The coefficient of each explanatory variable in the best model for each tick/pathogen species is represented with the size of dot. All the continuous explanatory variables were standardized. The p-value for each coefficient is represented as follows: \*\*\*, <0.001; \*\*, <0.01; \*, <0.05. The detail outputs from GLM are represented in Supplementary S6 Table.

suitable habitats for *I. persulcatus* were entirely encompassed by those for *I. ovatus*. The suitable habitats for *Haemaphysalis* ticks tended to be in the southern part of the study area, while those for *H. flava* were dispersed. Some grids at the northern mountains were predicted to have a high presence probability of *H. japonica* despite containing multiple absent sites of this species (Fig 2 and 4). The high mountainous regions (cf *elev* in Supplementary S2 Fig) were generally predicted to be suitable for pathogens. LDB had suitable habitats largely overlapped with *I. ovatus* and *I. persulcatus*, while some suitable habitats for *I. persulcatus* (i.e., some grids at the northern mountains) were not suitable for pLDB. A wide range of the southern mountains was predicted to be suitable for RFB. The suitable habitats for TBEV were at the northern mountains. BJNV had suitable habitats at both northern and southern mountains. YEZV showed scattered suitable habitats.

To quantify spatial overlap between pathogen and tick species, we calculated Schoener's *D* and Warren's *I* (Table 4). TBEV, YEZV, BJNV, pLDB, and RFB had the biggest Schoener's *D* and Warren's *I* with *I. persulcatus* and the second biggest values with *I. ovatus*. The values for these pathogens were not close to 1. Schoener's *D* and Warren's *I* of LDB with *I. ovatus* and *I. persulcatus* were close to 1.

## Discussion

Understanding the distribution of tick-borne pathogen and its determinants is crucial for public health protection. Many studies have tried to reveal the risk distribution of pathogen infection using data of disease occurrence in humans [50], vector tick records [51], and specific antibody in wild animals [52] obtained over long periods. Although the distribution of ticks infected with pathogens is also crucial for understanding the potential of infection [53], the low prevalence of tick-borne pathogens, particularly tick-borne viruses, in ticks poses a major challenge of spatial modeling [51]. In the present study, the short-term intensive tick collection followed by molecular detection of pathogens enables us to obtain enough comparable datasets of both ticks and pathogens in ticks for predicting their suitable habitats. Interestingly, the predicted

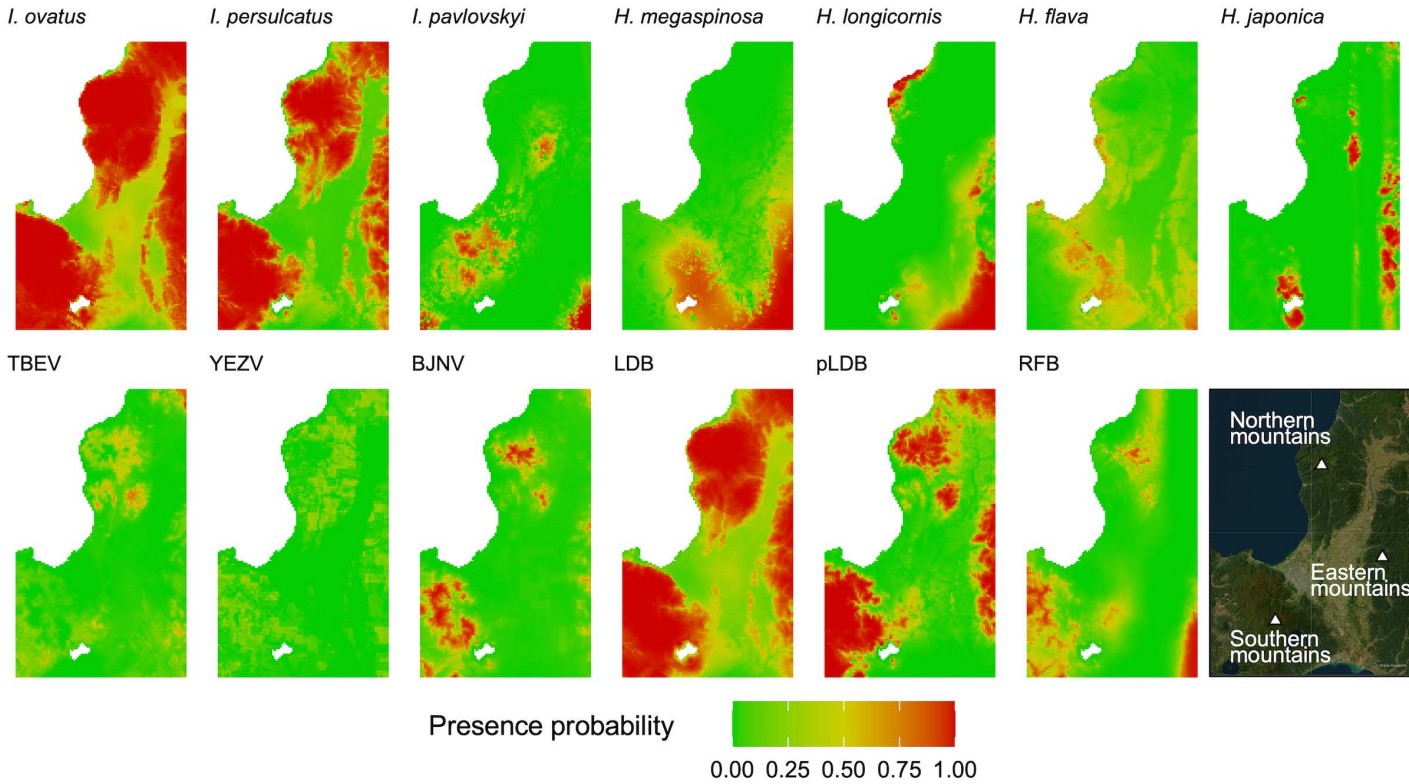

**Fig 4. Probability of species presence.** The presence probability of each tick/pathogen species was predicted by the best model and mapped with a resolution of 1 × 1 km. The satellite image on the bottom right is also shown in Fig 1 and 2. Basemap image was provided as an Esri's ArcGIS basemap. Content is the intellectual property of Esri and is used herein with permission. Copy right © 2025 Esri and its licensors. All rights reserved.

suitable habitats of tick-borne pathogens were different from those of their vector tick species; Schoener's *D* and Warren's *I* were not close to 1, except for those between LDB and *I. ovatus*/*I. persulcatus*. Our findings emphasize that the distribution of vector tick species only partially represents the distribution of pathogen-infected ticks and the local distributions of ticks infected with endemic pathogens should be comprehensively investigated in each region.

Through consideration of eighteen environmental variables, our ENM revealed that landscape, climatic, and topographic variables affect the distributions of a variety of tick species and pathogens. One of the climatic variables, yearly snow depth has a consistent influence on the distribution of ticks. The positive effect of snow depth on *Ixodes* ticks can result from the increased overwinter survival, as snow accumulation protects them from extreme cold events [54]. Besides, the negative effect of snow depth on the life cycle of *Haemaphysalis* ticks has been reported in previous reports in Japan [28,55]; snow depth decreases the abundance of sika deer, which is a crucial host for *Haemaphysalis* spp. ticks [7,56]. In contrast, the yearly snow depth was not included in the best models of TBEV, YEZV, BJNV, pLDB, and RFB. It was not significant in the single regression analyses for TBEV, YEZV, and RFB either (data not shown). These findings suggested that interactions with snow depth can cause the differences in distribution between ticks and pathogens in our study area. Among the factors considered, the distributions of wild mammals were rarely influential to either ticks or tick-borne pathogens. One possible reason for this is the spatial and temporal mismatches between our sampling data and the environmental predictors. The publicly available data of wild mammal distributions in the study area, which was presence/absence in 5 × 5 km grids, might fail to capture the fine-scale variations of mammal density. Up-to-date, high-resolution density data is desired for further investigation of interactions between mammal species and tick/pathogen species.

**Table 4. Schoener's *D* and Warren's *I* between each pathogen and tick.**

| | TBEV | | YEZV | | BJNV | | LDB | | pLDB | | RFB | |
|---|---|---|---|---|---|---|---|---|---|---|---|---|
| | *D* | *I* | *D* | *I* | *D* | *I* | *D* | *I* | *D* | *I* | *D* | *I* |
| *I. ovatus* | 0.500 (0.494 - 0.505)* | 0.794 (0.790 - 0.797) | 0.478 (0.473 - 0.484) | 0.762 (0.758 - 0.766) | 0.454 (0.449 - 0.459) | 0.792 (0.789 - 0.795) | 0.917 (0.916 - 0.919) | 0.994 (0.994 - 0.995) | 0.615 (0.609 - 0.620) | 0.867 (0.864 - 0.870) | 0.469 (0.463 - 0.475) | 0.756 (0.751 - 0.760) |
| *I. persulcatus* | 0.552 (0.546 - 0.559) | 0.828 (0.824 - 0.832) | 0.554 (0.548 - 0.559) | 0.836 (0.832 - 0.839) | 0.532 (0.527 - 0.537) | 0.844 (0.841 - 0.846) | 0.867 (0.863 - 0.870) | 0.978 (0.977 - 0.979) | 0.740 (0.734 - 0.746) | 0.928 (0.925 - 0.930) | 0.523 (0.516 - 0.530) | 0.785 (0.780 - 0.790) |
| *I. pavlovskyi* | 0.253 (0.243 - 0.264) | 0.486 (0.474 - 0.498) | 0.288 (0.276 - 0.299) | 0.512 (0.499 - 0.524) | 0.342 (0.330 - 0.355) | 0.569 (0.557 - 0.582) | 0.289 (0.283 - 0.296) | 0.563 (0.556 - 0.570) | 0.319 (0.309 - 0.328) | 0.552 (0.540 - 0.563) | 0.373 (0.360 - 0.386) | 0.617 (0.605 - 0.629) |
| *H. megaspinosa* | 0.297 (0.286 - 0.307) | 0.527 (0.517 - 0.537) | 0.260 (0.250 - 0.269) | 0.470 (0.460 - 0.481) | 0.226 (0.219 - 0.233) | 0.498 (0.490 - 0.507) | 0.390 (0.383 - 0.398) | 0.670 (0.664 - 0.676) | 0.326 (0.317 - 0.336) | 0.534 (0.524 - 0.544) | 0.307 (0.296 - 0.318) | 0.522 (0.512 - 0.532) |
| *H. longicornis* | 0.255 (0.243 - 0.268) | 0.464 (0.452 - 0.477) | 0.154 (0.144 - 0.164) | 0.291 (0.280 - 0.302) | 0.118 (0.112 - 0.125) | 0.302 (0.294 - 0.312) | 0.235 (0.228 - 0.242) | 0.474 (0.466 - 0.482) | 0.160 (0.152 - 0.168) | 0.322 (0.310 - 0.333) | 0.218 (0.204 - 0.234) | 0.351 (0.336 - 0.366) |
| *H. flava* | 0.365 (0.357 - 0.374) | 0.642 (0.634 - 0.650) | 0.324 (0.316 - 0.332) | 0.599 (0.592 - 0.606) | 0.272 (0.265 - 0.278) | 0.573 (0.566 - 0.581) | 0.533 (0.527 - 0.539) | 0.824 (0.820 - 0.827) | 0.353 (0.346 - 0.361) | 0.625 (0.618 - 0.633) | 0.337 (0.328 - 0.346) | 0.603 (0.594 - 0.611) |
| *H. japonica* | 0.259 (0.246 - 0.272) | 0.466 (0.452 - 0.481) | 0.197 (0.187 - 0.208) | 0.402 (0.390 - 0.415) | 0.210 (0.198 - 0.223) | 0.426 (0.412 - 0.440) | 0.222 (0.215 - 0.228) | 0.479 (0.472 - 0.487) | 0.255 (0.246 - 0.265) | 0.489 (0.477 - 0.500) | 0.176 (0.165 - 0.187) | 0.343 (0.329 - 0.357) |

*Values in parenthesis are 95% confidence intervals calculated using a bootstrap with 1,000 replicates.

In our short-term tick collection, *Ixodes* spp. was the most widely observed tick species across the sampling sites in central Hokkaido and was tested positive frequently with tick-borne pathogens compared with *Haemaphysalis* ticks. Furthermore, the results of Schoener's *D* and Warren's *I* demonstrated that the spatial distributions of all the pathogens are overlapped primarily with *I. persulcatus* and *I. ovatus*. In Hokkaido, human tick bites are largely caused by adult *I. persulcatus* [24,25], which was also found on human patients infected with YEZV and RFB [15,57]. Our findings, together with previous studies, suggest that adult *I. persulcatus* is the significant vector to transmit tick-borne pathogens to humans in this region. Besides, our results imply that *I. ovatus* is more abundant than *I. persulcatus* in the field, and thus, more *I. ovatus* than *I. persulcatus* are infected with TBEV or YEZV. Actually, all TBEV isolated from tick samples in Hokkaido were from *I. ovatus* [30,58,59]. Therefore, because of their roles in the natural circulation of these pathogens [29], not only *I. persulcatus* but also *I. ovatus* should be carefully monitored for preventing disease occurrence as well as further spreading of the pathogens.

Moran's *I* test and local spatial statistics suggested that distributions of almost all tick/pathogen species were spatially clustered in local to regional scale (a scale of tens of kilometers in radius), while *H. flava* and YEZV were suggested to be spatially dispersed compared to other tick or pathogen species, respectively; this result is also observed in the potential distribution predicted by ENM. Generally, species are spatially clustered due to the limited dispersal ability. Ticks and tick-borne pathogens almost completely depend on wild animals for their dispersal, and thus, host preference should have a great impact on their distribution patterns. Indeed, *H. flava* is more frequently collected from birds than other *Haemaphysalis* species [8]. In contrast to *H. flava*, *H. japonica* appears to be restricted from dispersing between suitable habitats, as shown that this tick species was not collected at some regions predicted to be suitable for it (e.g., southwestern seaside area of the northern mountains). As in the case of *H. flava*, scattered distribution of YEZV can be explained by frequent introduction by birds. Nishino et al. (2024) reported that eight isolates of YEZV were detected in a total of 2,534 ticks

collected from migratory birds in Hokkaido [9], while TBEV was not detected in the same samples (personal communication with the author). In addition, microfoci spatially stable for decades have been reported for TBEV [53], while an emergence of microfocus in a short period has been reported for YEZV [29]. Although more quantitative investigations on their preference for birds are needed, our results support that infection risk of YEZV is spatially sporadic compared to TBEV.

The present study highlighted the effectiveness of a short-term intensive tick collection with constant effort followed by ecological niche modeling. Besides, our study has some limitations. First, tick collection by a single 30-minute flagging per site in 1.5 months may risk false absences of ticks and pathogens, although the sampling period (13 May to 26 June, 2024) was set based on the phenological data of major tick species reported in the previous studies [24,28]. Second, while the Ct value cutoff (<45) for the RT-qPCR and qPCR assays was intended to maximize detection sensitivity, it introduced a potential for false positives. We could not entirely rule out false positives; confirmatory sequencing was only partially successful across the different pathogen species. Third, to effectively utilize the small number of presences for some pathogen species with the extremely low prevalence (i.e., TBEV and YEZV), LOOCV and a stratified 5-fold cross-validation were chosen in modeling; this choice may be criticized as spatially optimistic. Implementing spatially blocked cross-validation with expanded datasets, alongside the integration of other presence data (e.g., disease occurrence), will be necessary in future studies to mitigate spatial optimism and establish more reliable distribution predictions [53]. Despite these limitations, we are confident that the findings in the present study will be crucial basic information for ecological studies of ticks and tick-borne pathogens and also for establishing countermeasures against tick-borne diseases in the study area.

## Conclusions

In the present study, we performed tick collection at 171 sites in central Hokkaido in a short period (approximately 1.5 months) in 2024, followed by screening the endemic tick-borne pathogens and spatial analyses. Our results revealed the potential distributions of seven tick species and five tick-borne pathogens in the study area. *Ixodes persulcatus* and *I. ovatus* were identified as the primary ticks for determining the distribution of the pathogens. However, the spatial distribution of pathogens, except for LDB, was not equivalent to tick distribution. Snow depth seemed to form the differences in the predicted suitable habitats between ticks and pathogens. Frequent introduction by birds may contribute to the scattered distributions of *H. flava* and YEZV compared to other ticks and pathogens. Although the predictions should be interpreted with caution due to the low viral prevalences and inherent limitations in the model-validation process, our results can help in prioritizing areas for further surveillance. The present findings expand our understanding of the spatial distribution of ticks and tick-borne pathogens and ecological factors associated with them, highlighting the effectiveness of intensive tick sampling in a short period of time.

## Supporting information

**S1 Fig. Spatial weight to calculate Moran's *I*.** An element of spatial weights matrix (*w* in y axis), which is defined by the distance (km) between sites (*d* in x axis), was simulated by varying the parameter controlling the distance decay (*r* = 0.1, 1, 10, or 20).
(TIFF)

**S2 Fig. Distributions of explanatory variables.** Distribution of each environmental variable was mapped across the study area.
(TIFF)

**S3 Fig. Local spatial analysis to visualize hotspots.** Z-values of each tick and pathogen for G-star local spatial statistics using different distance decay (*r* = 1 or *r* = 10) was indicated with different color: red, z-value > 2.58; orange, z-value > 1.96; light blue, z-value < −1.96; blue, z-value < −2.58; grey, −1.96 ≤ z-value ≤ 1.96.
(TIFF)

**S4 Fig. Predicted suitable habitats in binary format.** The presence probability was converted to presence/absence using the threshold which maximize TSS. Grids predicted to be present are indicated in orange, while grids predicted to be absent are indicated in grey.
(TIFF)

**S1 Table. The number of collected ticks and pathogen-positive sample at each site.**
(XLSX)

**S2 Table. Information of tick pools and result of pathogen screening.** For positive samples, the mean Ct values are provided.
(XLSX)

**S3 Table. Present sites of ticks and tick-borne pathogens.** A present site of each tick species was defined as a site where one or more adult ticks were collected, and a present site of each pathogen species was defined as a site where one or more pools were tested positive.
(DOCX)

**S4 Table. Full results of Moran's $I$ test to investigate spatial clustering of each tick/pathogen species.** Moran's $I$ and p-value were calculated with the parameter controlling the distance decay ($r = 0.1$ to $r = 20$). The results using $r = 1$ or $r = 10$ are also shown in Table 3.
(XLSX)

**S5 Table. Validity of the best model for each tick/pathogen species.** AUC and TSS were calculated using leave-one-out cross-validation (LOOCV) or stratified 5-fold cross-validation. Results of Moran's $I$ test using $r = 1$ for the residuals of best model are shown in this table.
(DOCX)

**S6 Table. Outputs from the best GLM of each tick/pathogen species.** Coefficients and p-values of environmental variables are also visualized in Fig. 3.
(DOCX)

**S7 Table. Data for modeling.** Number of collected adult ticks, presence (1)/absence (0) of pathogen, and scaled environmental covariates at each site are provided with site coordinates anonymized.
(XLSX)

**S1 Text. Additional confirmation of RT-qPCR/qPCR positive samples.**
(DOCX)

## Acknowledgments

We thank Dr. H. Kawabata, Dr. T. Takano, and Dr. S. Kobayashi for kindly providing qPCR positive control for TBEV, LDB, and RFB. We also appreciate Dr. R. Nakao, Dr. N. Isoda, Dr. Y. Itakura, and Dr. K. Tabata for supporting the present study.

## Author contributions

**Conceptualization:** Mebuki Ito, Keita Matsuno.

**Data curation:** Mebuki Ito.

**Formal analysis:** Mebuki Ito.

**Funding acquisition:** Mebuki Ito, Yuma Ohari, Keita Matsuno.

**Investigation:** Mebuki Ito.

**Methodology:** Mebuki Ito, Yuma Ohari, Mai Kishimoto, Keita Matsuno.

**Project administration:** Keita Matsuno.

**Resources:** Mebuki Ito, Mai Kishimoto, Keita Matsuno.

**Software:** Mebuki Ito.

**Supervision:** Keita Matsuno.

**Validation:** Mebuki Ito.

**Visualization:** Mebuki Ito.

**Writing – original draft:** Mebuki Ito.

**Writing – review & editing:** Keita Matsuno.

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
