## [Decision Letter · Decision Letter 0]

3 Oct 2025

PONE-D-25-45949Spatial distribution of ticks and tick-borne pathogens in central Hokkaido, Japan and associated ecological factors  revealed by intensive short-term survey in 2024PLOS ONE

Dear Dr. Matsuno,

Thank you for submitting your manuscript to PLOS ONE. After careful consideration, we feel that it has merit but does not fully meet PLOS ONE’s publication criteria as it currently stands. Therefore, we invite you to submit a revised version of the manuscript that addresses the points raised during the review process.

We look forward to receiving your revised manuscript.

Kind regards,

Bersissa Kumsa, DVM, MSc, PhD

Academic Editor

PLOS ONE

3. We note that Figures 1, 2 and 4 in your submission contain [map/satellite] images which may be copyrighted. All PLOS content is published under the Creative Commons Attribution License (CC BY 4.0), which means that the manuscript, images, and Supporting Information files will be freely available online, and any third party is permitted to access, download, copy, distribute, and use these materials in any way, even commercially, with proper attribution. For these reasons, we cannot publish previously copyrighted maps or satellite images created using proprietary data, such as Google software (Google Maps, Street View, and Earth). For more information, see our copyright guidelines: http://journals.plos.org/plosone/s/licenses-and-copyright.

1. You may seek permission from the original copyright holder of Figures 1, 2 and 4 to publish the content specifically under the CC BY 4.0 license.

4. We note that there is identifying data in the Supporting Information file <TableS1>. Due to the inclusion of these potentially identifying data, we have removed this file from your file inventory. Prior to sharing human research participant data, authors should consult with an ethics committee to ensure data are shared in accordance with participant consent and all applicable local laws.

Data sharing should never compromise participant privacy. It is therefore not appropriate to publicly share personally identifiable data on human research participants. The following are examples of data that should not be shared.

-Location data

Please remove or anonymize all personal, ensure that the data shared are in accordance with participant consent, and re-upload a fully anonymized data set. Please note that spreadsheet columns with personal information must be removed and not hidden as all hidden columns will appear in the published file.

5. We notice that your supplementary figure 1 is uploaded with the file type 'Figure'. Please amend the file type to 'Supporting Information'. Please ensure that each Supporting Information file has a legend listed in the manuscript after the references list.

Additional Editor Comments:

Dear Authors,

The reviewers have completed their evaluation of your manuscript. I encourage you to revise and resubmit your work, ensuring that all reviewer comments are thoroughly addressed. Please incorporate the feedback carefully and provide a detailed, point-by-point response that clearly outlines every change made in response to the reviewers’ suggestions.

In addition, kindly correct all typographical and grammatical errors, and ensure that the manuscript is prepared in full compliance with the journal’s formatting and submission guidelines.

We look forward to receiving your revised submission.

Reviewers' comments:

Reviewer's Responses to Questions

**Comments to the Author**

1. Is the manuscript technically sound, and do the data support the conclusions?

Reviewer #1: Yes

Reviewer #2: Yes

Reviewer #3: Partly

2. Has the statistical analysis been performed appropriately and rigorously? 

Reviewer #1: Yes

Reviewer #2: I Don't Know

Reviewer #3: No

3. Have the authors made all data underlying the findings in their manuscript fully available?

Reviewer #1: No

Reviewer #2: Yes

Reviewer #3: No

4. Is the manuscript presented in an intelligible fashion and written in standard English?

Reviewer #1: Yes

Reviewer #2: No

Reviewer #3: Yes

5. Review Comments to the Author

Reviewer #1: Summary

This study conducted a short, intensive field survey (171 sites; 13 May–26 June 2024) in central Hokkaido, Japan, flagging for ticks under constant effort, pooling specimens, and screening for five pathogens. Using GLM-based ecological niche models, the authors predicted habitat suitability for seven tick species and the pathogens, checked spatial clustering via Moran’s I, applied LOOCV to validate, and quantified pathogen vector spatial overlap.

Introduction

The introduction section is too general; introduce specific tick species, relevant pathogens, and their public health relevance earlier.

Add a proper gap statement: prior Hokkaido work has patchy coverage, and this study contributes tick, pathogen data, and comparative ENMs.

Methods

Strong on geography and effort standardization. However, using a single collector and a fixed 30-minute effort per site, during a short period of 43 days, could bias detectability, so acknowledge this as a limitation.

Following Moran’s I, conducting Hotspot analysis & Cluster outlier analysis will point out the regions of hotspots, and add confidence to it.

Results

Provide AUC/TSS distributions (median, IQR) across species

Discussion

In discussion, don’t overstate “risk” as it creates conflicts with modeled suitability for human incidence. Please define outputs as the presence probability of infected ticks, because human behavior exposure, seasonality across the year, wasn’t measured, which limits how directly results translate to cases.

Close the discussion section with two or three concrete next steps and a single sentence “so what” message, so the Discussion ends with a clear take-home.

Data availability

Consider releasing data, full raster outputs, and code to enable reproducibility while protecting sites.

Line 136-138: More information on pools, how many pools in total, and species or genus-wise?

L144–198: Pooling rules are clear; add median pool sizes per species/stage and max Ct acceptance rationale. How non-amplifying pools were handled.

Line 221-223: Moran’s I being -1 should be interpreted as negative spatial autocorrelation with neighbors being very different from each other.

L226–291: GLM with LOOCV is transparent, but presence/absence derived from short-term sampling risks false absences, and LOOCV can be spatially optimistic. At least add a sensitivity analysis. Explain the VIF threshold and any variable interactions.

L226–256: There’s a scale mismatch (mammals 5×5 km, others 1×1 km). Discuss potential smoothing bias or add a sensitivity analysis without mammal layers.

Reviewer #2: The publication by Matsuno et al. attempts to define the precise spatial distribution of two tick genera, Ixodes and Haemaphysalis, and six potentially pathogenic microorganisms associated with ticks (three viruses and two bacteria) on Hokkaido Island (Japan). The authors study the potential impact of certain environmental factors (landscape, climate, topography) and wildlife.

Overall, the English needs to be revised to make the manuscript easier to read.

Some concepts need to be clarified and certain points need to be improved:

Introduction:

• The concept of a reservoir—both for ticks (lines 47–50) and for hosts (lines 54–58)—is never clearly mentioned, yet it is essential for understanding the circulation of potentially pathogenic microorganisms. This should be defined in the introduction.

• Lines 80–81: In Europe and the USA, nymphs are generally considered the highest risk. Why would adults be more at risk in Japan?

• Line 95: Consider removing the adjectives “intensive” and “comprehensive” and let the reader form their own judgment.

• Tick collection method: Typically, tick density is measured per unit area (e.g., per 100 m²) rather than per unit time. Why did the authors choose a 30-minute duration, which seems short? What approximate distance was covered during this time?

• Tick analysis method: It appears that pools were used to test both nymphs and adults, especially for virus detection in ticks. For bacteria, a pool of 100 nymphs is extremely large. While this is explained later in the text, the “Materials and Methods” section should specify that only presence/absence was ultimately measured.

• Lines 193–198: It is not clear how the ticks were genotyped for Borrelia. How were the Borrelia species identified—by genotyping targeting a specific gene, or by sequencing?

• Lines 200–204: This section is unclear. Please rephrase for clarity.

• Many abbreviations are used without being defined (e.g., pLDB on line 198, and throughout the Ecological Niche Modeling section: prec6, temp6, elev, angl, etc.). While these are explained in Table 1, they should also be defined in the text. I recommend moving Table 1 to the supplementary data.

Results :

• Increase the resolution of Table 2 by enlarging the font size.

• Lines 378–380: Why were hosts not included in the analysis, given that they are essential for ticks, which are strictly hematophagous?

• Table 4: Increase the resolution of the table by enlarging the font size.

Discussion:

• Lines 431–439: It is unclear what is actually being discussed here. Please revise. Previous studies have used presence/absence data, as in the present study. Other studies on human cases and serology in wildlife are also important and should be addressed.

• Lines 452–453: It would be interesting to develop this argument further. Why does snow cover impact tick distribution? Why would Ixodes be more affected by this factor compared to Haemaphysalis?

• Lines 460–461: I am truly surprised by this observation. Wildlife plays an essential role in the presence and maintenance of ticks in a given environment. Please clarify and rephrase.

• Lines 483–484: I understand that ticks infected with viruses may be clustered, but this is generally not the case for Lyme Borrelia and relapsing fever Borrelia. Please clarify.

Reviewer #3: The study addresses an important topic in vector ecology and provides novel data on tick and pathogen distributions in Hokkaido. The sampling effort is extensive and the integration of field data with modeling is timely. The manuscript is generally well-organized and written. However, several methodological and reporting issues need clarification or improvement before publication. Below I evaluate the scientific rigor, novelty, data interpretation, and clarity, and provide detailed comments on strengths, weaknesses, and revisions.

Scientific Rigor and Methodology

• Sampling Design: The authors sampled adult and nymphal ticks at 171 sites from 13 May–26 June 2024 using a standardized 30-minute flagging protocol by a single collector. This consistent, high-effort sampling across a broad area is a strength, yielding 4,608 adult and 5,300 nymph ticks from 159 sites (Table 2). The sample size is impressive and likely captures much of the regional tick diversity in spring. However, the short study period covers only late spring–early summer; tick phenology in Hokkaido may extend into autumn, and different species or life stages (e.g. larvae) might be active later. The authors should acknowledge this seasonal limitation. Site selection is not described in detail: how were the 171 locations chosen (e.g. random, stratified by habitat)? Clarification of site selection would strengthen confidence that sampling was representative and not biased toward easily accessible areas.

• Pathogen Detection: Molecular screening used published RT-qPCR/qPCR assays for each pathogen. The authors provide primer/probe details and thermal cycling conditions (Methods). Using published, specific assays (e.g. Achazi et al. 2011 for TBEV, Nishimura et al. 2022 for Yezo virus) is sound. One concern is the high Ct cutoff (<45) for positives; this may risk false positives at the very end of amplification. It would help if the authors commented on assay sensitivity/specificity (for example, whether positive controls and negative controls were included) or if any PCR products were confirmed by sequencing to rule out non-specific amplification. Including standard validation (e.g. replicates, sequencing representative amplicons) would strengthen confidence in the results.

• Statistical Modeling (Ecological Niche Models): The methods section lacks detail on the ENM procedure. It is not specified how absences or background data were chosen (e.g. were non-detection sites treated as absences?), nor which software or R packages were used. The variable selection process is unclear – did they start with all 18 covariates and use stepwise selection? Were collinearity or variance inflation assessed? The brief mention of applying “spatial smoothing” for some models is not explained. For reproducibility and rigor, the authors should provide more information: the modeling algorithm (e.g. GLM with binomial link), criteria for choosing best-fit models, and any preprocessing (e.g. standardizing variables, assessing correlations). As noted in the literature, SDMs for ticks require careful standardization and reporting. The manuscript would benefit from clarifying these steps. If possible, sharing the model code or parameters (e.g. in a supplementary file or repository) would help transparency.

A major concern is the small number of positive samples for some pathogens. For example, only 7 pools were positive for TBEV and 8 for Yezo virus. Fitting a reliable habitat model on such few presences is challenging and may lead to overfitting. The authors should discuss this limitation. In particular, note that with many zeros (true negatives) and few positives, standard logistic regression may produce unstable coefficients. Did the authors aggregate pathogens into genus-level (e.g. “all Borrelia”) or model each pathogen separately? It appears each pathogen was modeled individually (Table 4 shows separate D/I values), which could be under-powered for rarer pathogens. More discussion of model confidence is warranted. Despite this, the authors do report model performance metrics, which suggests at least some predictive signal.

• Environmental Covariates: The inclusion of habitat (forest, grassland), topography (elevation), climate (temperature, snow depth) and host presence is comprehensive. The finding that annual maximum snow depth strongly influenced tick distributions (positively for Ixodes ticks, negatively for Haemaphysalis) is plausible given Hokkaido’s climate. However, snow depth did not enter pathogen models. The manuscript suggests this may indicate complex “interactions” between snow and pathogen cycles. This interpretation is reasonable but a bit speculative; it might reflect that pathogens depend on host factors not captured by snow alone. The negligible effect of wild mammal distributions in models is likely due to the coarse (5×5 km presence/absence) host data. The authors correctly note that finer-scale host data are needed to assess these effects. Overall, the choice and handling of covariates seems sound, but more detail on how each was processed (e.g. how snow depth was measured or averaged) should be added.

• Statistical Analysis: Beyond ENMs, the authors calculated Schoener’s D and Warren’s I to quantify niche overlap between each pathogen and each tick species (Table 4). This is appropriate for comparing spatial distributions. They also computed Moran’s I for clustering. These analyses are straightforward and correctly interpreted (e.g. low overlap D between most viruses and ticks implies distinct distributions). Their results that I. persulcatus and I. ovatus have the highest overlap with all pathogens (D,I up to ~0.95 for LDB) are consistent with prevalence data (see below). Statistical methods are adequately applied, but the manuscript should ensure that all reported p-values and thresholds are explained (e.g. what is a “significant” Moran’s I).

Data Interpretation and Conclusions

The authors’ interpretation of results is generally sound and well-supported by the data. Overall, the conclusions follow logically from the results. One area for improvement is to explicitly acknowledge the limitations of pathogen sample sizes when interpreting model outputs. For instance, predictions for TBEV and Yezo virus (each <10 positives) should be treated with caution. But the authors generally provide a balanced discussion, citing relevant literature (e.g. stable TBEV foci vs emergent YEZV foci) to contextualize their findings.

In summary, I find the study potentially suitable for publication after major revision. The authors should clarify their modeling methods, ensure compliance with data-sharing policies, and address the points listed above. With these revisions, the manuscript would represent a strong contribution to tick-borne disease ecology.

6. PLOS authors have the option to publish the peer review history of their article (what does this mean?). If published, this will include your full peer review and any attached files.

Reviewer #1: No

Reviewer #2: **Yes:** Nathalie BOULANGER

Reviewer #3: No

---

## [Author Response · Author response to Decision Letter 1]

6 Nov 2025

Response to the comment of Academic editor:

The reviewers have completed their evaluation of your manuscript. I encourage you to revise and resubmit your work, ensuring that all reviewer comments are thoroughly addressed. Please incorporate the feedback carefully and provide a detailed, point-by-point response that clearly outlines every change made in response to the reviewers’ suggestions.

In addition, kindly correct all typographical and grammatical errors, and ensure that the manuscript is prepared in full compliance with the journal’s formatting and submission guidelines.

We look forward to receiving your revised submission.

Response: We appreciate the editor’s contribution to our manuscript. We have completed the revision of the manuscript and response to the reviewer’s comment as follows. In addition to the revisions incorporating feedback by the reviewers, we carefully checked the submission guidelines of your journal to revise our manuscript. Typographical and grammatical errors were also corrected.

Response to the comments of Reviewer #1:

1. The introduction section is too general; introduce specific tick species, relevant pathogens, and their public health relevance earlier. Add a proper gap statement: prior Hokkaido work has patchy coverage, and this study contributes tick, pathogen data, and comparative ENMs.

Response: We revised the third paragraph in introduction section to explain the detail situations of Hokkaido, including specific tick species and their relevant pathogens, and to clearly state a knowledge gap (lines 86-95). Besides, for responding to the other reviewers’ suggestions, we retained the first and second paragraph for broad readers of the journal.

2. Strong on geography and effort standardization. However, using a single collector and a fixed 30-minute effort per site, during a short period of 43 days, could bias detectability, so acknowledge this as a limitation.

Response: We agree with this comment and mentioned it as one of the limitations of the present study in the discussion section (lines 589-592).

3. Following Moran’s I, conducting Hotspot analysis & Cluster outlier analysis will point out the regions of hotspots, and add confidence to it.

Response: Thank you for your suggestion to strengthen our spatial analyses. Following the reviewer’s suggestion, we additionally calculated the local spatial statistic G-star to point out hotspots (lines 244-249). The result was described in result section (on lines 431-435) and z-sores were visualized in Supplementary Fig. S3. The caption for Fig. S3 is on lines 833-837. The pointed out hotspots generally aligned with the mapped raw data on Fig. 2.

4. Provide AUC/TSS distributions (median, IQR) across species

Response: The median and IQR of AUC and TSS across species were provided on lines 442-444.

5. In discussion, don’t overstate “risk” as it creates conflicts with modeled suitability for human incidence. Please define outputs as the presence probability of infected ticks, because human behavior exposure, seasonality across the year, wasn’t measured, which limits how directly results translate to cases.

Response: We followed this suggestion and revised the sentences lines 509-510, 513-516, and 522-525 to erase the word “risk.” In addition to the reviewer’s suggestion, we also referred to comment 24 by reviewer #2 and revised the first paragraph of discussion section (lines 510-516) to improve clarity.

6. Close the discussion section with two or three concrete next steps and a single sentence “so what” message, so the Discussion ends with a clear take-home.

Response: We appreciate the reviewer’s suggestion to improve our manuscript. We added one paragraph at the end of discussion section, where the future prospects were described (lines 600-606).

7. Consider releasing data, full raster outputs, and code to enable reproducibility while protecting sites.

Response: All the explanatory variables in ecological niche modeling were gained from open data, which readers can freely obtain from each recourse. The response variables in the ecological niche modeling and data for spatial analysis, except for their coordinates, were shown in Supplementary Table S1, which was mistakenly removed from the initial submission. In this study, our variable selection include an iterative manual process where we made decisions at each step based on a simultaneous assessment of multiple metrics (AUC, TSS, VIFs, and Moran’s I). Therefore, to ensure the greatest clarity, we have provided a detailed, step-by-step description of the criteria and the selection process in the method section (lines 304-339). We believe this detailed description is more informative and transparent than a script of the manual selection.

8. Line 136-138: More information on pools, how many pools in total, and species or genus-wise?

Response: Since we thought that only the pooling strategy should be mentioned in the materials and methods section, the actual information of pools has been described in the results section. The number of pools in total (1026 samples) was explained in the second paragraph of the result section (lines 377-378) and species/genus-level information was summarized in Table 2. The medians of pooled individual ticks per species/genus were added in Table 2 to follow the reviewer’s comment 9.

9. L144–198: Pooling rules are clear; add median pool sizes per species/stage and max Ct acceptance rationale. How non-amplifying pools were handled.

Response: Following the reviewer’s suggestion, we added the medians of pooled individual ticks per species/genus in Table 2.

The max Ct acceptance (< 45) was to maximize the sensitivity of our screening, aiming to capture as many potential positives as possible for our ecological niche modeling. Since we believe the genetic detections are the most sensitive methods, negative samples in each RT-qPCR/qPCR were treated as negative. Regarding the specificity of the screening methods, we performed additional examinations including sequencing of fragments. Please kindly refer to the detail results of our additional examinations provided in our response to the comment 29 by the reviewer #3. We described the rationale and validation results as a limitation of our study on lines 592-597.

10. Line 221-223: Moran’s I being -1 should be interpreted as negative spatial autocorrelation with neighbors being very different from each other.

Response: Thank you for reading the detail of our manuscript. According to the reviewer’s suggestion, the interpretation of Moran’s I value was revised as follows: “Moran’s I close to –1 means that neighboring sites are different from each other, while Moran’s I close to 1 means that the species distribute in cluster.” on lines 240-242.

11. L226–291: GLM with LOOCV is transparent, but presence/absence derived from short-term sampling risks false absences, and LOOCV can be spatially optimistic. At least add a sensitivity analysis. Explain the VIF threshold and any variable interactions.

Response: Following the reviewer’s suggestion to add a sensitivity analysis, we additionally performed a stratified 5-fold cross-validation to evaluate the stability of the model's predictive performance. The methods were described on lines 339-344 and the results were explained on lines 444-449 and supplementary Table S4. The means of AUC and TSS calculated by stratified 5-fold cross-validation were above acceptable thresholds for predictions (AUC > 0.7 and TSS > 0.5) and their standard deviations were small, except for H. flava. We agree with the reviewer’s concern associated with our sampling and modeling methods, and the limitations were described in discussion section (lines 589-592 and 597-602).

Explanations on VIF and variable interactions were added on lines 326-329 and 308-310. In short, VIF threshold was set at 5 and no interaction terms of environmental variables were considered in order to avoid complexity in modeling.

12. L226–256: There’s a scale mismatch (mammals 5×5 km, others 1×1 km). Discuss potential smoothing bias or add a sensitivity analysis without mammal layers.

Response: For the mammal variables, the data from each 5×5 km grid was assigned uniformly to all 25 of the 1×1 km grids nested within it (i.e., a block assignment was performed instead of smoothing). We revised the sentence to explain the assignment method of mammal distribution on lines 294-295. We also acknowledge the reviewer’s concern that this approach assumes a uniform distribution of the mammal species within each 5×5 km grid. The limitations associated with the scale mismatch was added in the discussion section (on lines 542-547).

Response to the comments of Reviewer #2:

13. The concept of a reservoir—both for ticks (lines 47–50) and for hosts (lines 54–58)—is never clearly mentioned, yet it is essential for understanding the circulation of potentially pathogenic microorganisms. This should be defined in the introduction.

Response: We appreciate the reviewer’s advice to improve our manuscript. Following the reviewer’s suggestion, we revised the sentences to clearly use the word “reservoir” on lines 47-49 and 54-56.

14. Lines 80–81: In Europe and the USA, nymphs are generally considered the highest risk. Why would adults be more at risk in Japan?

Response: We understand the reviewer’s wonder at the regional difference. At least, local epidemiological data from Hokkaido show that the most tick bites are caused by adult ticks; Miyamoto et al. (1991) reported that all of 58 tick bites were caused by adults and Sasaki et al. (2021) reported that only one out of 153 tick bites was caused by nymphal tick. Based on this local situation that adult ticks are the main vector for humans in this region, our research was designed to focus on them. We revised the sentence on lines 86-89 in introduction section to more clearly emphasize this specific epidemiological situation in our study area.

15. Line 95: Consider removing the adjectives “intensive” and “comprehensive” and let the reader form their own judgment.

Response: According to the reviewer’s suggestion, we revised the sentence to erase the abstract words and to make the concrete explanation in introduction section (lines 98-100).

16. Tick collection method: Typically, tick density is measured per unit area (e.g., per 100 m²) rather than per unit time. Why did the authors choose a 30-minute duration, which seems short? What approximate distance was covered during this time?

Response: Indeed, we don’t aim to determine tick density in the present study. Because of the topographical diversity in our study area, we were not able to set an area in which a person could freely move for flagging. Furthermore, we prioritized increasing the spatial coverage across the study area by sampling at a larger number of sites. We selected a time-based sampling since it is more robust and convenient to standardize the sampling effort in our study area. In response to your question, the approximate distance covered during the 30-minute period was 750 meters. We added the sentence on lines 141-143 to explain the reason why we selected a 30-minute duration for flagging. We also acknowledge the reviewer’s concern that a 30-minute flagging was short and might miss minor tick species at each site, and described this limitation in discussion section (lines 589-592).

17. Tick analysis method: It appears that pools were used to test both nymphs and adults, especially for virus detection in ticks. For bacteria, a pool of 100 nymphs is extremely large. While this is explained later in the text, the “Materials and Methods” section should specify that only presence/absence was ultimately measured.

Response: We added a sentence on lines 217-219 in materials and methods section to emphasize the data of pathogen screening was used as presence/absence data in the subsequent spatial analyses. For readability, the sub-sections “Nucleic acid extraction and pathogen detection” and “Definition of presence/absence of each tick/pathogen at a site” were retained separately.

18. Lines 193–198: It is not clear how the ticks were genotyped for Borrelia. How were the Borrelia species identified—by genotyping targeting a specific gene, or by sequencing?

Response: We agree that detailed genotyping of Borrelia is important for molecular epidemiological studies as performed in multiple studies from our study area (Nakao et al., 1994; Masuzawa, 2004; IDSC, 2011; Takano et al, 2014; Okado et al., 2021), which were also cited in the manuscript. The aim of our study was not to reproduce the species specificity of individual genotypes, but rather to understand the spatial distribution of the major species groups. To achieve this, we used the multiplex qPCR assay from Takano et al. (2014), which is well-validated for specifically distinguishing Borrelia burgdorferi s.l. (LDB) from B. miyamotoi (RFB). Since the previous studies cited above have indicated that LDB specifically carried by I. persulcatus are causing Lyme disease in our study area, we treated LDB detected in I. persulcatus (referred as pLDB) as a separate variable during the statistical analyses in addition to the total LDB. We revised the sentences on lines 205-213 to clarified this rationale.

19. Lines 200–204: This section is unclear. Please rephrase for clarity.

Response: We added a sentence in this section for clearly explaining the definition to convert the data of tick collection and pathogen detection to presence/absence data for the subsequent statistical analyses (lines 217-219).

20. Many abbreviations are used without being defined (e.g., pLDB on line 198, and throughout the Ecological Niche Modeling section: prec6, temp6, elev, angl, etc.). While these are explained in Table 1, they should also be defined in the text. I recommend moving Table 1 to the supplementary data.

Response: Thank you for your comment. We spelled out all the abbreviations pointed out by the reviewer at the first appearance in the text (lines 211-213, 254-256, 261-265, and 287-291). For easier understanding of modeling method, we retained Table 1 in the main manuscript.

21. Increase the resolution of Table 2 by enlarging the font size.

Response: The font size of Table 2 was enlarged following the reviewer’s comment. We also attached an Excel file named “Table2&4forReview”, which includes Table 2 for efficient review process.

22. Lines 378–380: Why were hosts not included in the analysis, given that they are essential for ticks, which are strictly hematophagous?

Response: All the tick species appeared in the manuscript are known as hematophagous. We further discussed the reasons why mammal variables were rarely included in the best model (lines 542-547). Please also refer to the response to your comment 26, where we explain the reason in other words.

23. Table 4: Increase the resolution of the table by enlarging the font size.

Response: The font size of Table 4 was enlarged following the reviewer’s comment. Please also find an attached Excel file named “Table2&4forReview”, which contains 4 for efficient review process.

24. Lines 431–439: It is unclear what is actually being discussed here. Please revise. Previous studies have used presence/absence data, as in the present study. Other studies on human cases and serology in wildlife are also important and should be addressed.

Response: We agree with the reviewer’s comment that occurrences of diseases and serological data from wildlife are also important. In this section, we want to claim that short-term intensive tick collections can be used for revealing the local distributions of ticks infected with endemic pathogens. This information, in turn, should help to understand infection risks from a different perspective than disease and serological data. We revised the sentences on lines 510-516 to clarify that we can use the data of ticks infected with pathogens as well as other data for revealing the spatial distribution of infection risk.

25. Lines 452–453: It would be interesting to develop this argument further. Why does snow cover impact tick distribution? Why would Ixodes be more affected by this factor compared to Haemaphysalis?

Response: We added the discussion on the interaction between snow and tic

---

## [Decision Letter · Decision Letter 1]

25 Feb 2026

PONE-D-25-45949R1Spatial distribution of ticks and tick-borne pathogens in central Hokkaido, Japan and associated ecological factors  revealed by intensive short-term survey in 2024PLOS One

Dear Dr. Matsuno,

Thank you for submitting your manuscript to PLOS ONE. After careful consideration, we feel that it has merit but does not fully meet PLOS ONE’s publication criteria as it currently stands. Therefore, we invite you to submit a revised version of the manuscript that addresses the points raised during the review process.

We look forward to receiving your revised manuscript.

Kind regards,

Stephen M. Rich, MS, PhD

Academic Editor

PLOS One

Journal Requirements:

**Additional Editor Comments:**

In particular, Reviewer #3 has included several comments suggestions in the annotated manuscript.  Please reply to those concerns.

Reviewers' comments:

Reviewer's Responses to Questions

**Comments to the Author**

1. If the authors have adequately addressed your comments raised in a previous round of review and you feel that this manuscript is now acceptable for publication, you may indicate that here to bypass the “Comments to the Author” section, enter your conflict of interest statement in the “Confidential to Editor” section, and submit your "Accept" recommendation.

Reviewer #1: All comments have been addressed

Reviewer #2: All comments have been addressed

Reviewer #3: All comments have been addressed

2. Is the manuscript technically sound, and do the data support the conclusions?

Reviewer #1: Yes

Reviewer #2: Yes

Reviewer #3: No

3. Has the statistical analysis been performed appropriately and rigorously? 

Reviewer #1: Yes

Reviewer #2: Yes

Reviewer #3: No

4. Have the authors made all data underlying the findings in their manuscript fully available?

Reviewer #1: Yes

Reviewer #2: Yes

Reviewer #3: No

5. Is the manuscript presented in an intelligible fashion and written in standard English?

Reviewer #1: Yes

Reviewer #2: Yes

Reviewer #3: No

6. Review Comments to the Author

Reviewer #1: The authors have made substantial revisions in response to the reviewers’ comments, and I am satisfied that the manuscript has been significantly improved. I recommend this manuscript for publication.

Reviewer #2: (No Response)

Reviewer #3: The manuscript presents valuable data and a well‑structured ecological modeling framework; however, several important methodological, statistical, and reporting issues remain and require substantial revision before the study can meet PLOS ONE standards. Detailed, line‑by‑line and section‑by‑section comments have been provided in the attached document, including notes on sampling design limitations, PCR confirmation procedures, spatial cross‑validation, model interpretation, data availability, and English‑language clarity. Please address all points in the attached review document in your next revision.

7. PLOS authors have the option to publish the peer review history of their article (what does this mean?). If published, this will include your full peer review and any attached files.

Reviewer #1: No

Reviewer #2: No

Reviewer #3: **Yes:** Makwarela Tsireledzo Goodiwill

---

## [Author Response · Author response to Decision Letter 2]

1 Mar 2026

Point-by-point responses to reviewer comments

We really thank the consideration by editor and reviewers for our manuscript. All changes to the manuscript are marked by “Track Changes” option in 'Revised Manuscript with Track Changes'. Please find our response to your comments below, where the line numbers indicate the line numbers in the clean copy without track changes ('Manuscript').

Response to the comment of Academic editor:

1. In particular, Reviewer #3 has included several comments suggestions in the annotated manuscript. Please reply to those concerns.

Response: Thank you for providing the opportunity to revise our manuscript. In accordance with Reviewer #3’s comments and the suggestions in the annotated manuscript, we have thoroughly revised the manuscript. We provide point-by-point justifications in this file for the comments for which we consider a detailed explanation is necessary. All other minor changes (e.g. grammatical corrections) requested in the annotated file have also been incorporated and replied to all of the reviewer #3’s comments in the annotated file (“Manuscript_replied.pdf”).

Response to the comments of Reviewer #3:

2. The manuscript presents valuable data and a well‑structured ecological modeling framework; however, several important methodological, statistical, and reporting issues remain and require substantial revision before the study can meet PLOS ONE standards. Detailed, line‑by‑line and section‑by‑section comments have been provided in the attached document, including notes on sampling design limitations, PCR confirmation procedures, spatial cross‑validation, model interpretation, data availability, and English‑language clarity. Please address all points in the attached review document in your next revision..

Response: Thank you for your tremendous contribution to our manuscript. We greatly appreciate the time and effort you invested in providing detailed, line-by-line suggestions in the annotated document. We have carefully revised the manuscript in accordance with the comments and replied to each of your comments individually within the attached file (“Manuscript_replied.pdf”). For the major comments requiring clarification, our detailed justifications are also provided below.

3. Add one sentence of scope/results:

Response: We agree with your suggestion to bridge the methodology and the results here. To achieve this, we have added a sentence describing our results of tick collection and pathogen detection and modified the above/following sentences on lines 29-35 to fit in the context. The revised sentences summarize the scope of our collection, initial screening results, and modeling outcome.

4. Missing controls: time‑of‑day, weather/ground moisture, vegetation height/density—all affect catchability. Please report these covariates or acknowledge as sampling bias.

Response: According to your suggestion, we have now specified the sampling conditions in the Materials and Methods section (on lines 141-142 and 144-147). Specifically, we clarify that we collected ticks during daylight hours on days with no precipitation across various habitat types. In addition, the sentence in Discussion to mention our limitation of sampling was also revised (lines 613-616).

5. One 30‑min pass per site risks false absences. Consider an occupancy/detection sensitivity paragraph in the Discussion.

Response: We revised the last paragraph in the Discussion to incorporate this limitation together with the above your comment #4 (line 613-616). Since this issue is closely associated with pathogen detection method and our modeling approach, we integrated this discussion into the existing paragraph to provide a more cohesive explanation of the study's limitations, rather than creating a separate section.

6. “Samples with Ct<45 and exponential curves were considered positive”; partial sequencing only; no technical replicate reporting; no stated negative‑control outcomes. Action: Report (i) replicate policy, (ii) negative‑control results, (iii) confirmation rate by sequencing per pathogen, and (iv) GenBank accession IDs where available. This is essential given several positives at Ct > 39.

Response: Thank you for pointing out these essential details. To address your concerns, we have revised the sentences in the Materials and Methods (lines 204–211) to explicitly state that all assays were performed in technical replicate with positive and no-template negative controls, and no amplification was observed in any negative controls. Furthermore, we have provided Supplementary Text S1, where further confirmation by alternative PCR assay, sequence data, and detailed justifications are provided. We have now cited this Supplemental information on line 395-397 in the Result section.

7. “Tick present if ≥1 adult collected; pathogen present if ≥1 positive pool.” This excludes nymphs from tick presence despite being numerous and ecologically relevant. Please justify (morphological limits are fine) and discuss consequences for ENM.

Response: We used this definition to ensure the highest accuracy of morphological identification. The sentence was added on lines 231-232 to explain this point. In addition, the sentence to explain the definition (on lines 229-231) were revised for readability.

8. Provide the rationale for the reported r values in Table 3 (you show r=1 and r=10) and a short justification for choosing these for emphasis. Local G*: You computed and mapped Z‑scores; consider summarizing notable clusters in the main text (currently in Supplement).

Response: We selected r = 1 and r = 10 for Table 3 to represent the lower and upper bounds of our sensitivity analysis, respectively. These two values effectively capture the overall trend of the results (Supplementary Table S4). We revised the footnote of Table 3 (lines 443-446) to justify our choice.

As suggested, we have revised the paragraph describing the results of the local spatial statistic (Lines 449–452) to highlight notable clusters. This revision provides a clearer summary of the spatial risks within the main text, reducing the need for constant reference to the Supplementary information.

9. Random CV (LOOCV, 5‑fold) can be spatially optimistic. Please add a spatially blocked CV sensitivity analysis (e.g., k‑fold by spatial clusters or spatial buffering) to re‑estimate AUC/TSS, at least in Supplement. Covariates: You considered 20 variables (+quadratics) from JAXA LULC (10 m), MLIT climate/topography (1 km), and mammals (5 km presence/absence). You then downscaled mammal layers uniformly across 1 km cells, which introduces pseudo‑precision; the manuscript should state this limitation clearly (you do briefly in Discussion—bring a sentence into Methods). Interactions: None were considered; add one sentence justifying this (to limit complexity and preserve power at N=171 sites).

Response: We appreciate the reviewer’s insightful suggestion regarding the potential for spatial optimism in our cross-validation. While we agree that spatially blocked CV is a robust approach, implementing it effectively with our data was impossible with convergence. Therefore, rather than presenting unstable results, we have explicitly acknowledged this as a limitation in the Discussion (lines 625-628). We revised the text to emphasize that future studies with expanded datasets should employ spatially blocked CV and integrate additional occurrence data to establish more reliable predictions.

As for this possible pseudo‑precision of mammal distribution, we have added a sentence to the Methods section (lines 304–306) to mention it as follows: “Although the 5-km grid is the highest resolution available for mammal distribution data, uniform downscaling to 1 km might overlook fine-scale variations.”

We also revised the sentence on lines 319-321 to justify no interaction terms according to the suggestion.

10. consider adding bootstrap CIs for these indices as a robustness check.

Response: We appreciate the reviewer’s comment to provides a more robust assessment in our analysis. As suggested, 95% confidence intervals were calculated for both Schoener’s D and Warren’s I using a bootstrap approach with 1,000 replicates. The method was described on line 373-375 and the updated results were added in Table 4.

11. Note that your mammal layers are coarse, and snow data are long‑term means; explicitly state these temporal and spatial mismatches.

Response: Thank you for pointing out these data limitations. We have revised the sentences (on lines 565–571) incorporating the phrase “spatial and temporal mismatches” to acknowledge the constraints of our environmental predictors.

12. Concise and faithful. Add one sentence on surveillance application (e.g., grids with high suitability for I. persulcatus overlapping with TBEV/pLDB/RFB are priorities), while reiterating low viral prevalence and model‑validation caveats.

Response: We appreciate your thoughtful comment to improve our conclusion. We added a sentence according to the reviewer as follows (lines 644-647): “Although the predictions should be interpreted with caution due to the low viral prevalence and inherent limitations in the model-validation process, our results can help in prioritizing areas for future surveillance.”

13. Provide a public dataset sufficient to reproduce the analysis, e.g.:1×1 km grid centroids (or jittered points within grid cells) for each site;per‑site tick counts (by species/sex/stage), pool metadata (pool size, Ct values, assay), and pathogen outcomes;processed environmental covariates or scripts to fetch and process them;R code for GLM/GAM, Moran’s I/G*, overlap metrics, and mapping. Deposit to Zenodo/Dryad/GitHub with DOIs and update the statement accordingly.

Response: We fully agree with the importance of data transparency and reproducibility. However, due to concerns regarding potential reputational damage to the local communities and specific locations surveyed, we are unable to publicly release the exact or jittered GPS coordinates of the sampling sites. We have addressed your request by providing the following alternative datasets and resources:

First, the results of tick collection and pathogen screening at each site are provided in Supplementary Table S1. We have also added Supplementary Table S2, which provides the pool size and mean Ct values of positive samples, and cited it on line 395.

Furthermore, processed environmental covariates are provided in Supplementary Table S7, which was cited in Data Availability section. Finally, we have deposited the representative R code on GitHub (https://github.com/DRAM-IIZC/MebukiITO_Riskmap).

The Data Availability section was updated accordingly (lines 651-657).

---

## [Editor Report · Decision Letter 2]

20 Apr 2026

PONE-D-25-45949R2

Spatial distribution of ticks and tick-borne pathogens in central Hokkaido, Japan and associated ecological factors revealed by intensive short-term survey in 2024PLOS One

Dear Dr. Matsuno,

Thank you for submitting your manuscript to PLOS ONE. After careful consideration, we feel that it has merit but does not fully meet PLOS ONE’s publication criteria as it currently stands. Therefore, we invite you to submit a revised version of the manuscript that addresses the points raised during the review process.

Please see the annotated list of edits/corrections that I have compiled.  Please make these changes in your manuscript. Note that throughout the manuscript, articles ('the,' 'a,' 'an') are occasionally missing, particularly before singular countable nouns. This is common in writing by non-native English speakers and should be checked systematically (I may have missed some instances). Also, while passive voice is acceptable in scientific writing, some passages become difficult to follow due to excessive passive constructions. Consider revising key sentences to active voice for clarity.

We look forward to receiving your revised manuscript.

Kind regards,

Stephen M. Rich, MS, PhD

Academic Editor

PLOS One

Journal Requirements:

Additional Editor Comments:

This is a well-structured ecological study with valuable data on tick and tick-borne pathogen distributions. The methodology is sound and the findings are scientifically significant. However, the manuscript still requires attention to grammar, usage, and clarity—particularly subject-verb agreement, article usage, and some awkward phrasings. Below is a list of issues organized by location.

Title and Short Title

Line 15 (Short title): "tic k-borne" → "tick-borne" (spurious space)

Abstract

Lines 35–38: The phrase "the predicted suitable habitats were specific to each pathogen/tick species" is somewhat awkward. Consider: "the predicted suitable habitats differed among pathogen and tick species."

Introduction

Line 49–50:"infect to humans and domestic animals" → "infect humans and domestic animals" (delete "to")

Line 50–51:"are reservoirs of the tick-borne pathogens, which are responsible for harboring them for long period" → "are reservoirs of tick-borne pathogens, harboring them for long periods" (delete "the"; "period" → "periods")

Line 57–58:"not only amplifying pathogens to transmit to ticks" → "not only by amplifying pathogens for transmission to ticks" (clearer construction)

Line 60:"understanding on associations among" → "understanding of the associations among"

Line 65:"In example" → "For example"

Line 74:"the circulation of tick-borne pathogens are indirectly affected" → "the circulation of tick-borne pathogens is indirectly affected" (subject-verb agreement: "circulation" is singular)

Line 79:"climatic and biological characters" → "climatic and biological characteristics"

Line 88:"while BJNV distributes widely" → "while BJNV is distributed widely" or "while BJNV is widely distributed"

Methods

Tick sampling

Line 140:"Study period was set from" → "The study period was set from" (add article)

Line 137–138:"To confirm that the investigated sites randomly located across" → "To confirm that the investigated sites were randomly located across" (add "were")

Definition of presence/absence

Lines 229–231:"pathogen present if ≥1 positive pool; the others sites were absent sites" → "pathogen present if ≥1 positive pool; all other sites were classified as absent" (clearer; "others" → "other")

Line 231:"for this definition for tick to ensure" → "for this definition for ticks to ensure" (plural)

Spatial clustering

Line 257:"Z-vales" → "Z-values" (typo)

Ecological niche modeling

Line 313:"Ecological niche modelling was performed" — Note inconsistent spelling: "modelling" (British) vs. "modeling" (American) used elsewhere. Choose one and be consistent throughout.

Line 331:"a manual forward and backward stepwise" → "a manual forward and backward stepwise procedure" or "manual forward and backward stepwise selection"

Line 337:"performance package [45].When" → "performance package [45]. When" (add space after period)

Line 356:"Finaly" → "Finally" (typo)

Results

Prevalence section

Line 381:"suggested that the 171 investigated sites located randomly" → "suggested that the 171 investigated sites were located randomly"

Table 3

Line 439:"Supplementary Table 3" → Should this be "Supplementary Table S4"? (inconsistent with other references)

Model performance section

Line 483–484:"(Moran's I test, p > 0.05) was detected... (Supplementary Table S5)." — The period after the parenthetical creates a fragment. Revise: "No significant spatial autocorrelation (Moran's I test, p > 0.05) was detected in the residuals of the final models (Supplementary Table S5)."

Predicted suitable habitats

Line 503:"predicted to have a high presence probability of H. japonica despite containing multiple absent sites of these species" → "...of this species" (H. japonica is singular)

Line 510:"BJNV have suitable habitats" → "BJNV has suitable habitats" (subject-verb agreement)

Discussion

Lines 533–534:"There are many studies tried to reveal" → "Many studies have tried to reveal"

Line 546:"the distribution of vector tick species only partially represent" → "the distribution of vector tick species only partially represents" (subject-verb agreement)

Line 552:"the yearly snow depth has a consistent influence" → Consider: "yearly snow depth has a consistent influence"(article usage with abstract quantity)

Line 554:"The positive effect of snow depth on Ixodes ticks can result from the increased overwinter survival by snow accumulation" → "The positive effect of snow depth on Ixodes ticks may result from increased overwinter survival, as snow accumulation protects them from..." (clearer causation)

Line 557:"the previous reports in Japan[28,55]" → "previous reports in Japan [28,55]" (add space before bracket; remove "the")

Line 557:"snow depth decreases abundance of sika deer" → "snow depth decreases the abundance of sika deer"

Line 561:"It was not significant in the single regression for TBEV, YEZV, and RFB, neither" → "It was not significant in single regression analyses for TBEV, YEZV, or RFB either" ("neither" → "either" at end of negative clause; "analyses" for clarity)

Line 564:"rarely influential to either ticks and tick-borne pathogens" → "rarely influential for either ticks or tick-borne pathogens" ("and" → "or" with "either")

Line 569:"Uptodate" → "Up-to-date" or "Updated"

Line 583:"isolated form tick samples" → "isolated from tick samples" (typo: "form" → "from")

Line 593:"species is spatially clustered due to" → "species are spatially clustered due to" (or "a species is spatially clustered")

Line 596:"Actually, H. flava is more frequently collected" → "Indeed, H. flava is more frequently collected" ("Actually" is often considered informal)

Line 598–600:"H. japonica should be strictly prevented from moving between the suitable habitats" — This phrasing is unclear. Do you mean the tick's dispersal is limited? Consider: "H. japonica appears to be restricted from dispersing between suitable habitats"

Line 603:"Nishino et al. (2024) has reported" → "Nishino et al. (2024) reported" (simple past for citing literature; or "have reported" if treating as plural)

Line 619:"We could not be entirely ruled out false positives" → "We could not entirely rule out false positives"

Line 624:"this choice may not escape criticism for being spatially optimistic" — Consider: "this choice may be criticized as spatially optimistic" (more direct)

Conclusions

Line 640:"primary ticks to determining distribution" → "primary ticks for determining the distribution" or "primary ticks determining the distribution"

General comments

The word "respectively" appears frequently and is sometimes used incorrectly (when there is no clear one-to-one correspondence). Review each instance and retain only where genuinely matching paired items.

Throughout the manuscript, articles ("the," "a," "an") are occasionally missing, particularly before singular countable nouns. This is common in writing by non-native English speakers and should be checked systematically.

While passive voice is acceptable in scientific writing, some passages become difficult to follow due to excessive passive constructions. Consider revising key sentences to active voice for clarity.

Several instances of extra or missing spaces near punctuation and citations (e.g., "[28,55]" vs. " [28,55]"). Standardize throughout.

• "tick-borne" should always be hyphenated when used as a compound adjective

• "short-term" should be hyphenated when preceding a noun

---

## [Author Response · Author response to Decision Letter 3]

23 Apr 2026

Response to editor comments

We sincerely appreciate the editor’s efforts in providing point-by-point corrections to our manuscript. We have incorporated all the suggestions including the proper usage of articles. Furthermore, we have conducted a thorough grammatical review of the entire manuscript. Specifically, we removed the word "respectively" from several sentences to improve clarity and flow (Lines 167, 216, 399, 432, 544, and 584 in the clean version). We also revised some key sentences from passive voice to active voice (Lines 452-454, 550-552, and 636-638 in the clean version). All changes to the manuscript are marked by “Track Changes” option in 'Revised Manuscript with Track Changes'.

As for the comment “Line 15 (Short title): "tic k-borne" → "tick-borne" (spurious space)”, we have verified the Short Title and confirmed that "tick-borne" is correctly formatted without a space in our manuscript. We have re-saved the file to ensure this error does not appear in the final version.

Correction to the calculation of Warrant’s I:

We would like to inform the editor that we have corrected the formula used to calculate Warren’s I. In our original submission, we used the formula provided in the original article by Warren et al. (2008) [48]:

I(p_X,p_Y) =1-1/2 √(∑_i〖(√(p_(X,i) )-√(p_(Y,i) ))〗^2 ).

However, we identified that this specific formula contained an erratum, which was subsequently corrected by the authors in 2011 (https://academic.oup.com/evolut/article/65/4/1215/6854212). The correct formula is as follows:

I(p_X,p_Y) =1-1/2 ∑_i〖(√(p_(X,i) )-√(p_(Y,i) ))〗^2 .

We have accordingly updated the Methods (line 369) and the results in Table 4. Crucially, since the correct formula is a monotonic transformation of the original one, this revision does not alter the relative rankings of the I values and, therefore, does not affect our claim in the manuscript.

---

## [Editor Report · Decision Letter 3]

30 Apr 2026

Spatial distribution of ticks and tick-borne pathogens in central Hokkaido, Japan and associated ecological factors  revealed by intensive short-term survey in 2024

PONE-D-25-45949R3

Dear Dr. Matsuno,

We’re pleased to inform you that your manuscript has been judged scientifically suitable for publication and will be formally accepted for publication once it meets all outstanding technical requirements.

Kind regards,

Stephen M. Rich, MS, PhD

Academic Editor

PLOS One
---

## [Editor Report · Acceptance letter]

PONE-D-25-45949R3

PLOS One

Dear Dr. Matsuno,

I'm pleased to inform you that your manuscript has been deemed suitable for publication in PLOS One. Congratulations! Your manuscript is now being handed over to our production team.

Kind regards,

on behalf of

Dr. Stephen M. Rich

Academic Editor

PLOS One